# DiScoFormer: Plug-In Density and Score Estimation with Transformers

**Vasily Ilin** [1 2]  **Peter Sushko** [3]  **Ranjay Krishna** [4 3]

## Abstract

Estimating probability density and its score from samples remains a core problem in generative modeling, Bayesian inference, and kinetic theory. Existing methods are bifurcated: classical kernel density estimators (KDE) generalize across distributions but suffer from the curse of dimensionality, while neural score-matching models achieve high precision but require retraining for every target distribution. We introduce DiScoFormer (Density and Score Transformer), an equivariant Transformer that maps i.i.d. samples to both density values and score vectors. Unlike score matching, which learns a fixed function $\mathbb{R}^d \to \mathbb{R}^d$ for a single distribution, DiScoFormer learns a *sequence-to-sequence* operator that generalizes across distributions and sample sizes without retraining. Analytically, we prove that self-attention can recover normalized KDE, establishing it as a functional generalization of kernel methods; empirically, individual attention heads learn multi-scale, kernel-like behaviors. The model outperforms KDE for density and score estimation, and provides a plug-in score oracle for score-debiased KDE, Fisher information computation, and Fokker–Planck-type PDEs.

## 1. Introduction

Estimating a probability density $f$ and its corresponding score function $\nabla \log f$ from i.i.d. samples is a foundational problem in statistical inference, underpinning tasks in generative modeling, stochastic control, and information theory. While classical non-parametric methods, such as Kernel Density Estimation (KDE), offer strong theoretical guaran-

tees and respect fundamental symmetries, they are hampered by a rigid bias-variance trade-off and a prohibitive "curse of dimensionality" as dimension $d$ increases. Conversely, modern neural score-matching models achieve high accuracy in high-dimensional settings but are typically transductive: they must be retrained for every new target distribution, precluding their use as general-purpose, "off-the-shelf" estimators.

We propose bridging this gap by framing density and score estimation as a sequence-to-operator (Sutskever et al., 2014) learning task. Given an i.i.d. sample $X = \{x_i\}_{i=1}^n \subset \mathbb{R}^d$, we seek to learn a single model that maps the entire sample $X$ to its underlying density and score functions. Specifically, we define an operator $T$ for the log-density and $S$ for the score:

$$\begin{pmatrix} x_1 \\ \vdots \\ x_n \end{pmatrix} \xrightarrow{T} \begin{pmatrix} \log f(x_1) \\ \vdots \\ \log f(x_n) \end{pmatrix}, \quad \begin{pmatrix} x_1 \\ \vdots \\ x_n \end{pmatrix} \xrightarrow{S} \begin{pmatrix} \nabla \log f(x_1) \\ \vdots \\ \nabla \log f(x_n) \end{pmatrix}.$$

A critical requirement for such an operator is that it must respect the inherent symmetries of the data. Specifically, the estimates should be permutation-equivariant with respect to the sample index and affine-equivariant with respect to the coordinate space. For a permutation matrix $P$, invertible matrix $A$, and shift $\mu$, these symmetries are defined as:

$$T(PXA + \mathbf{1}\mu^\top) = P\,T(X) - \log|\det A|\,\mathbf{1},$$
$$S(PXA + \mathbf{1}\mu^\top) = P\,S(X)\,A^{-\top},$$

Building on these principles, we introduce DiScoFormer – Density and Score Transformer. By treating the sample $X$ as a sequence, the Transformer architecture provides permutation equivariance by construction. We achieve affine equivariance through a specialized whitening mechanism that normalizes inputs up to orthogonal transformations, combined with data augmentation to ensure rotation invariance. Unlike traditional models, our architecture utilizes cross-attention, allowing for the evaluation of $f$ and $\nabla \log f$ at arbitrary query points without retraining.

Furthermore, we provide a theoretical bridge between modern attention mechanisms and classical non-parametrics. We demonstrate analytically that under specific parameterizations, self-attention weights recover normalized KDE weights. This expressivity result positions the Transformer

[1]Department of Mathematics, University of Washington, Seattle, USA [2]Math AI Lab, University of Washington, Seattle, USA [3]Allen Institute for Artificial Intelligence, Seattle, USA [4]Paul G. Allen School of Computer Science & Engineering, University of Washington, Seattle, USA. Correspondence to: Vasily Ilin <vilin@uw.edu>.

*Proceedings of the 43$^{rd}$ International Conference on Machine Learning*, Seoul, South Korea. PMLR 306, 2026. Copyright 2026 by the author(s).

not as a black box, but as a principled, data-adaptive generalization of kernel methods that can dynamically learn multiscale smoothing behaviors from data.

Empirically, the model outperforms KDE and score-debiased KDE (Epstein et al., 2025) across a range of dimensions and sample sizes on both in-distribution and out-of-distribution test cases. Trained on Gaussian Mixture Model (GMM) data, the model generalizes to GMMs with more modes and non-Gaussian targets such as the Laplace distribution and demonstrates favorable scaling in both $n$ and $d$. By amortizing the estimation cost, our model provides high-fidelity plug-in oracles for downstream tasks, including entropy estimation, Fisher information calculation, and solving Fokker-Planck-type partial differential equations (PDEs).

In summary, our contributions are:

- DiScoFormer, a universal Transformer model for one-shot estimation of density and score functions from i.i.d. samples.

- A proof and empirical evidence establishing self-attention as a data-adaptive generalization of normalized kernel density estimation.

- Non-parametric density and score estimators that surpass classical KDE methods in accuracy across sample sizes and dimensions.

- Applications to Fisher information, differential entropy, and Fokker-Planck-type PDEs.

## 2. Related Work

Our work lies at the intersection of nonparametric density estimation, score-based methods, permutation-invariant neural architectures, and operator learning for probability measures.

**Kernel density estimation.** Classical nonparametric methods such as Parzen windows (Parzen, 1962) and Silverman's rule-of-thumb KDE (Silverman, 1986; Wand & Jones, 1994) estimate densities via local kernel averaging. While theoretically grounded, these methods suffer from a rigid bias–variance trade-off and scale poorly with dimension (Scott, 2015). Adaptive bandwidth methods (Abramson, 1982) and fast algorithms such as the Fast Gauss Transform (Greengard & Strain, 1991) partially alleviate computational costs but do not resolve the fundamental curse of dimensionality. Score-Debiased KDE (Epstein et al., 2025) demonstrates that access to a score oracle enables bias correction, motivating our approach: we provide precisely this missing oracle as a reusable, distribution-agnostic component.

**Other nonparametric density estimators.** $k$-nearest-neighbor methods (Loftsgaarden & Quesenberry, 1965), orthogonal series estimators (Efromovich, 2010; Scott, 1985), and normalizing flows (Rezende & Mohamed, 2015; Dinh et al., 2017; Papamakarios et al., 2017) offer alternatives to KDE but either lack direct score estimates or require per-distribution retraining. Our model provides both density and score from a single pretrained network.

**Score matching and score-based generative models.** Score matching (Hyvärinen, 2005) and its denoising variant (Vincent, 2011) estimate unnormalized model scores without computing partition functions. Sliced score matching (Song et al., 2019) scales these ideas to higher dimensions. Score-based generative models (Song & Ermon, 2019; Ho et al., 2020; Song et al., 2021b;a) learn scores along diffusion processes for sampling, with fast solvers (Lu et al., 2022; Karras et al., 2022) and convergence guarantees (Chen et al., 2023; 2024). These methods train on a single target density; in contrast, our model learns a universal score operator that generalizes across distributions without retraining.

**Permutation-invariant architectures.** DeepSets (Zaheer et al., 2017) and Set Transformers (Lee et al., 2019) provide foundational architectures for exchangeable data, while equivariant graph networks (Maron et al., 2019) extend these ideas to richer symmetry groups. Neural Processes (Garnelo et al., 2018; Edwards & Storkey, 2017) and Perceiver IO (Jaegle et al., 2022) learn distributions over functions from context sets. Recent work shows that permutation-equivariant transformers can universally approximate mean-field dynamics (Biswal et al., 2025). Our architecture exploits the same symmetry but outputs statistical quantities—density and score—while additionally enforcing affine equivariance.

**Attention as kernel smoothing.** Formal connections between softmax attention and kernel methods have been developed through linear-attention reformulations (Katharopoulos et al., 2020), random-feature approximations (Choromanski et al., 2021), Nyström methods (Xiong et al., 2021), explicit Gaussian-kernel replacements (Chen et al., 2021; Lu et al., 2021a), and Nadaraya–Watson regression interpretations (Zhang et al., 2023). Our Proposition 3.3 builds on this lineage by showing that (cross-)attention weights can exactly reproduce normalized Gaussian KDE weights at arbitrary query points.

**Neural operators.** Neural Operators (Kovachki et al., 2023), DeepONet (Lu et al., 2021b), and Fourier Neural Operators (Li et al., 2021) learn mappings between infinite-dimensional function spaces. The Score Neural Operator (Liao et al., 2024) extends this framework to probability distributions via RKHS embeddings. In contrast, our trans-

*Table 1.* Relative MSE of affine equivariance error, averaged over 50 trials.

| Transform | Rel. MSE |
| --- | --- |
| Permutation | 0 |
| Translation | 0 |
| Isotropic scaling | 0 |
| Anisotropic scaling | 0 |
| Rotation | $5 \times 10^{-4}$ |
| Full affine | $1 \times 10^{-4}$ |

former acts directly on raw i.i.d. samples without kernel embeddings, yielding a simpler operator that generalizes across densities and dimensions.

**Particle methods and Fokker–Planck solvers.** Interacting particle methods for Fokker–Planck equations (Carrillo et al., 2019; Maoutsa et al., 2020) and score-based transport modeling (Boffi & Vanden-Eijnden, 2023; Lu et al., 2024; Ilin et al., 2025) require accurate score estimates at each timestep. Stein Variational Gradient Descent (Liu & Wang, 2016; Liu, 2017; Duncan et al., 2023) similarly relies on kernel-based score approximations. Our model provides a fast, pretrained oracle that plugs directly into these solvers, avoiding per-distribution retraining.

## 3. Methodology

In this section we describe the core ideas of the proposed method: the symmetry-aware architecture, the relationship to classical kernel density estimation, the training pipeline, and implementation.

### 3.1. Equivariance

First, Proposition 3.1 establishes the equivariance properties of log-density and score estimation; here $T$ and $S$ map a sample to the log-density and score of its *underlying* density, so the transformed left-hand sides are evaluated for the pushforward law (see the Appendix). The proof is in the Appendix.

**Proposition 3.1** (Permutation and affine equivariance of log-density and score evaluation). *Let $f$ be a differentiable density, and $X = (x_1, \ldots, x_n)^T$ be its iid sample. Define*

$$T(X) := \begin{pmatrix} \log f(x_1) \\ \vdots \\ \log f(x_n) \end{pmatrix}, \quad S(X) := \begin{pmatrix} \nabla \log f(x_1)^\top \\ \vdots \\ \nabla \log f(x_n)^\top \end{pmatrix}.$$

*Let $P \in \mathbb{R}^{n \times n}$ be a permutation matrix, $A \in \mathbb{R}^{d \times d}$ be invertible, $\mu \in \mathbb{R}^d$, and $\mathbf{1} \in \mathbb{R}^n$ be the vector of ones. Then*

$$T\big(PXA + \mathbf{1}\mu^\top\big) = P\,T(X) - \log|\det A|\,\mathbf{1},$$
$$S\big(PXA + \mathbf{1}\mu^\top\big) = P\,S(X)\,A^{-\top}.$$

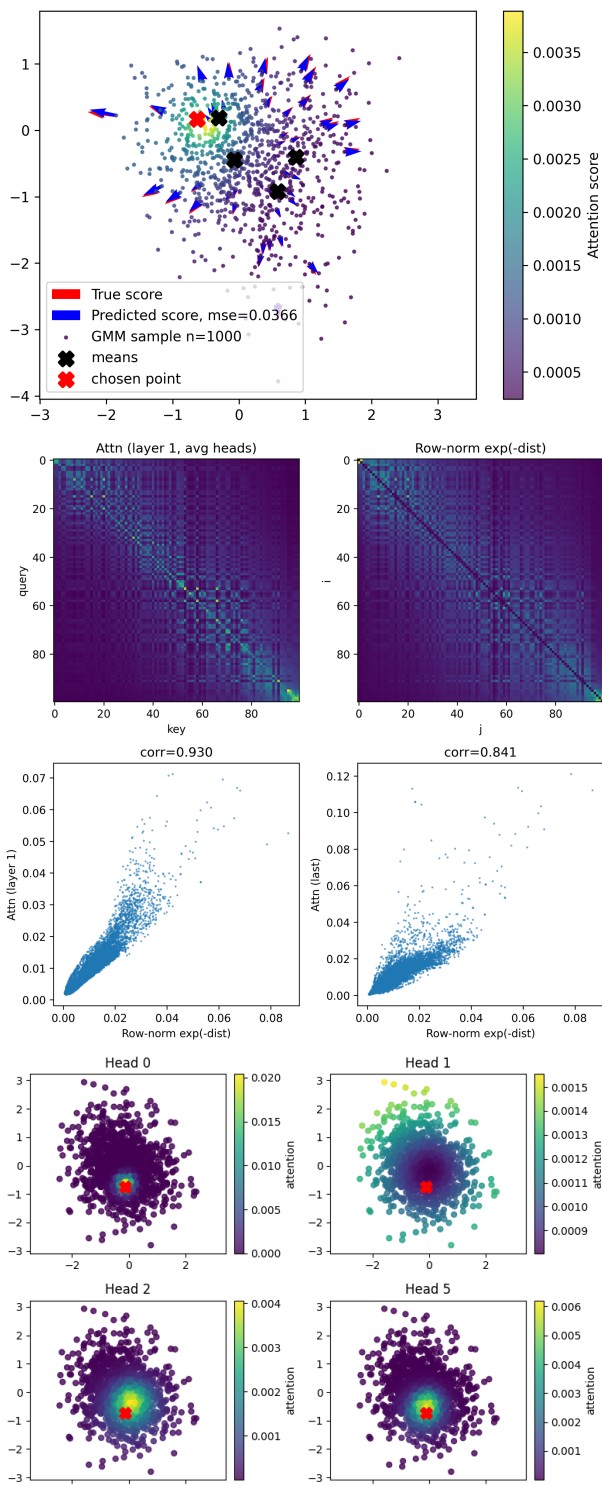

*Figure 1.* Attention visualization. Top: average attention in layer 0 from query x; heatmaps show the attention matrix and the normalized KDE matrix $D_{ij} \propto e^{-\|x_i - x_j\|_2^2}$, and scatter plots show very high agreement. Bottom: individual heads exhibit emergent specialization (close-range, far-range, directional).

```python
def forward(self, X, Y):
    m = X.mean(0, keepdim=True)
    Xc, Yc = X - m, Y - m
    S = Xc.T @ Xc + eps * torch.eye(d)
    A = matrix_sqrt_inv(S)
    Xw, Yw = Xc @ A, Yc @ A
    log_dens, score = self._core(Xw, Yw)
    log_dens += torch.logdet(A)
    return log_dens, score @ A.T
```

*Figure 2.* The forward pass implementing affine equivariance. Here `self._core` is a standard Transformer encoder without positional encodings, and `matrix_sqrt_inv(S)` computes the matrix inverse square root $S^{-1/2}$ (not element-wise). The core outputs the log-density and score in whitened coordinates; the log-density is shifted by $\log \det A = \log \det(S^{-1/2})$ to account for the change of variables, and the score is mapped back by $A^\top$.

To capture these symmetries, we use the Transformer architecture without positional encodings, combined with an affine normalization layer (Figure 2). The whitening step centers the data, decorrelates the features, and equalizes their variances by computing the inverse matrix square root of the regularized scatter matrix $S = X_c^\top X_c + \varepsilon I$. This transforms the input up to an arbitrary orthogonal (rotation/reflection) transformation in $O(d)$.

*Remark* 3.2 (Approximate affine equivariance). Whitening normalizes the sample covariance to the identity, so it handles *every* invertible linear map exactly up to a residual orthogonal ($O(d)$) transformation; together with centering, this means translations are handled exactly and any affine map is reduced to an $O(d)$ rotation/reflection. This residual is resolved approximately: training on randomly oriented GMMs encourages the network to learn approximately $O(d)$-invariant features, closing the gap in practice. Table 1 confirms this empirically – the relative MSE of the equivariance error under full affine transformations is on the order of $10^{-4}$.

### 3.2. Attention and KDE

We now establish a precise connection between the attention mechanism and Gaussian kernel smoothing. We state it for *cross*-attention, in which a set of query points $Y$ attends over a set of context points $X$; self-attention is the special case $Y = X$ (Corollary 3.6). Unlike prior formulations that require $L^2$-normalized inputs (Zhang et al., 2023), the following result holds for *arbitrary* input vectors. The proof follows from the polarization identity and is given in the Appendix.

**Proposition 3.3** (Cross-attention computes reweighted Gaussian kernel smoothing). *Let $X \in \mathbb{R}^{n_x \times d}$ with rows $x_1, \ldots, x_{n_x}$ (context/keys) and $Y \in \mathbb{R}^{n_y \times d}$ with rows $y_1, \ldots, y_{n_y}$ (queries). For any positive semi-definite $B \in$*

$\mathbb{R}^{d \times d}$, *define the cross-attention matrix*

$$A_{ij} = \frac{\exp(y_i^\top B x_j)}{\sum_{k=1}^{n_x} \exp(y_i^\top B x_k)}.$$

*Then*

$$A_{ij} = \frac{w_j \exp\left(-\frac{1}{2}\|y_i - x_j\|_B^2\right)}{\sum_{k=1}^{n_x} w_k \exp\left(-\frac{1}{2}\|y_i - x_k\|_B^2\right)},$$

*where $w_j = \exp\left(\frac{1}{2}\|x_j\|_B^2\right)$ and $\|z\|_B^2 = z^\top B z$.*

**Corollary 3.4** (Attention recovers KDE on constant-norm context). *If $\|x_j\|_B = c$ for all context tokens $j$ (e.g., $L^2$-normalized inputs with $B = h^{-2}I$), then $w_j$ is constant and cancels, giving $A_{ij} \propto \exp\left(-\|y_i - x_j\|_B^2/2\right)$, the normalized Gaussian kernel.*

Proposition 3.3 shows that a single attention head implements a *reweighted* Gaussian kernel, the reweighting $w_j = \exp\left(\frac{1}{2}\|x_j\|_B^2\right)$ being the only obstruction to exact KDE. We now show that supplying each token with a single scalar feature – its squared norm – removes this obstruction, so that one residual cross-attention block (model width $d_{\text{model}} \geq 2d + 1$), followed by an affine readout of its output *and* the per-query attention log-normalizer $\ell_i = \log \sum_j \exp(q_i^\top k_j)$, represents the *exact* classical KDE score and log-density at arbitrary query points. This makes the connection between the architecture and nonparametric statistics fully constructive. The proof is in the Appendix.

**Proposition 3.5** (Cross-attention represents the KDE score and log-density). *Fix $h > 0$ and let $K_h(y, x) = \exp(-\|y - x\|^2/(2h^2))$. Apply the fixed lift $z \mapsto [z, \|z\|^2]$ to every context token $x_j$ and query token $y_i$. A single residual cross-attention block (one head, exact softmax, no positional encodings, normalization, or feed-forward; model width $d_{\text{model}} \geq 2d + 1$), followed by an affine readout of the block output and the per-query log-normalizer $\ell_i := \log \sum_j \exp(q_i^\top k_j)$, maps any context $X \in \mathbb{R}^{n_x \times d}$ and queries $Y \in \mathbb{R}^{n_y \times d}$ to the exact KDE score and log-density of $\hat{f}_{h,X}$ at every query $y_i$:*

$$\nabla_y \log \hat{f}_{h,X}(y_i) = \frac{1}{h^2} \left( \frac{\sum_j K_h(y_i, x_j)\, x_j}{\sum_j K_h(y_i, x_j)} - y_i \right),$$

$$\log \hat{f}_{h,X}(y_i) = \ell_i - \frac{\|y_i\|^2}{2h^2} - \log n_x - \frac{d}{2}\log(2\pi h^2).$$

**Corollary 3.6** (Self-attention recovers KDE at the sample points). *Taking $Y = X$ in Propositions 3.3 and 3.5 recovers the corresponding self-attention statements: a single residual self-attention block over the lifted samples reproduces the normalized Gaussian KDE weights, the KDE score $\nabla \log \hat{f}_{h,X}(x_i)$, and the KDE density $\hat{f}_{h,X}(x_i)$.*

Together, Propositions 3.1–3.6 establish that the proposed Transformer architecture is not a black box applied to density estimation—it is structurally aligned with the task: attention can implement classical KDE.

---

**Algorithm 1** GMM DataLoader

---

**Input:** batch size $B$, dimension $d$, sample sizes $n_x, n_y$, components $[k_{\min}, k_{\max}]$

**repeat**

    Sample $k \in \{k_{\min}, \ldots, k_{\max}\}$

    **for** $b = 1$ **to** $B$ **do**

        Sample two random GMMs with $k$ components each

        Sample $X_b$ from the first, $Y_b$ from the second

        Compute $\log f_{X_b}(y)$ and $\nabla \log f_{X_b}(y)$ for $y \in Y_b$

    **end for**

    **Output:** $\left(X, Y, \log f_X(Y), \nabla \log f_X(Y)\right)$

**until** training stops

---

### 3.3. Empirical Verification

We verify whether the model learns kernel-like behavior in practice. To empirically verify the relationship between the KDE weights and attention scores, we visualize attention in Figure 1. As expected, the Transformer learns to attend to nearby points, with one attention head attending over far away samples instead. The attention mechanism effectively learns a kernel-like behavior, with emergent head specialization. The learned features allow for much better scaling both in the sample size and dimension, compared to KDE. In section C of the Appendix we further visualize the attention scores of individual heads, and find an emergent behavior of head specialization. Specifically, head 1 specialized to look at far-away points, whereas heads 0, 2, and 5 specialized to look at close- and mid-range interactions. Finally, heads 3, 4, 6, and 7 specialized to look in specific directions. This emergent behavior further links the multi-head Transformer to kernel-based methods, but with flexible multi-scale kernels.

### 3.4. Training Data

We train the Transformer to approximate the density and score of Gaussian Mixture Models (GMMs) from i.i.d. samples. GMMs are chosen because they are dense in the space of smooth densities (in total variation, and – with enough components – in Fisher divergence for the scores; see Appendix B.2), and both their densities and scores admit closed-form expressions. Training samples are generated on the fly (Algorithm 1). The objective combines the mean squared errors (MSE) of the predicted log-density and score through a convex weighting:

$$\mathcal{L}_T = \frac{1}{n} \|T(X, Y) - \log f_X(Y)\|_2^2,$$
$$\mathcal{L}_S = \frac{1}{n} \|S(X, Y) - \nabla \log f_X(Y)\|_2^2$$
$$\mathcal{L} = \alpha \mathcal{L}_T + (1 - \alpha)\mathcal{L}_S.$$

### 3.5. Model Variants and Implementation

We explore several architectural variants aimed at extending density and score estimation beyond the observed samples $X$, enabling evaluation at arbitrary query points $Y$. To achieve this, we introduce a cross-attention mechanism that processes two complementary inputs: a context tensor encoding the observed samples and a query tensor specifying the target locations. Through this interaction, the model learns to relate queries to the statistical structure of the observed data, producing accurate estimates of both $f_X(y)$ and $\nabla \log f_X(y)$ for any $y \in Y$. The cross-attention layer thus serves as a learned nonparametric smoother, effectively generalizing density and score estimation across the input domain rather than being limited to the training samples.

Recognizing that density and score estimation are mathematically coupled – linked through the gradient of $\log f$ – we employ a unified architecture with a shared backbone and two specialized output heads. This joint formulation encourages mutual consistency between the two quantities and allows the model to leverage shared geometric and statistical features during training, improving both accuracy and sample efficiency. It also enables test-time training: because the two heads must satisfy $S(C, Q)_i = \nabla_{q_i} T(C, Q)_i$ (the score is the gradient of the predicted log-density at the query $q_i$, with the context $C$ held fixed), their disagreement defines a label-free consistency loss that adapts the model to out-of-distribution inputs at inference, with no ground-truth density or score required (Section 4.1).

## 4. Experiments

We empirically investigate:

1. score and density estimation accuracy vs. KDE and SD-KDE (Epstein et al., 2025);

2. scaling in sample size $n$ and dimension $d$;

3. out-of-distribution generalization;

4. downstream use as a plug-in score oracle in SD-KDE, entropy/Fisher information estimation, and deterministic SBTM-like solvers for Fokker–Planck-type PDEs.

Since score matching methods such as DDPM and SDE-based models (Ho et al., 2020; Song et al., 2021b) learn a score for a single distribution and must be retrained per target, the closest method sharing our *nonparametric, plug-in* setting is KDE. We compare against sliced score matching in Appendix D and to multiple KDE bandwidth strategies (Scott, oracle, SD-KDE) throughout.

All experiments are performed on a single 48GB L40S GPU. The architecture is a permutation- and affine-equivariant

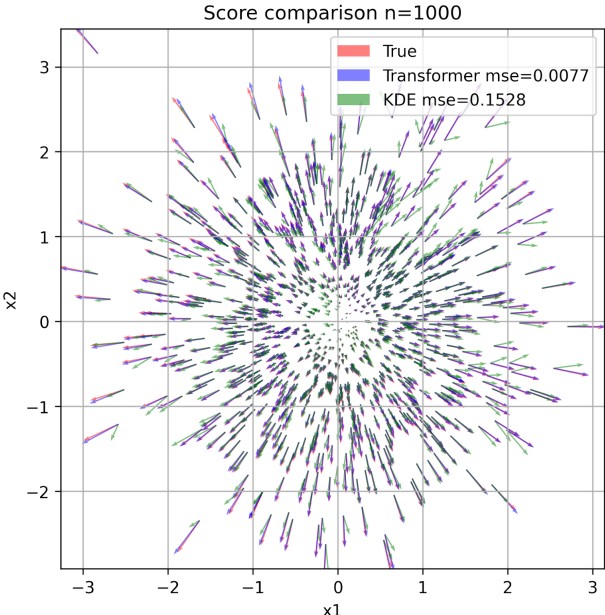

*Figure 3.* Score estimation comparison between Scott KDE (Scott, 2015) and our transformer model. The transformer is more accurate, especially in the sparse regions. We plot the negated score for easier viewing.

Transformer with 4 encoder layers (hidden size 128, 8 heads, GELU, pre-normalization), no positional encodings, and roughly 800,000 parameters. Unless stated otherwise we use batch size 32, sample size $n = 2048$, dropout 0.1, and draw GMMs with 1–10 modes with means in $[-3, 3]^d$ and diagonal covariances in $[0.2, 1]^d$. A runtime comparison with KDE is provided in Appendix E.

## 4.1. Score estimation

To visually compare our model to KDE, we first sample a 2d Gaussian, visualize the estimated scores using a quiver plot

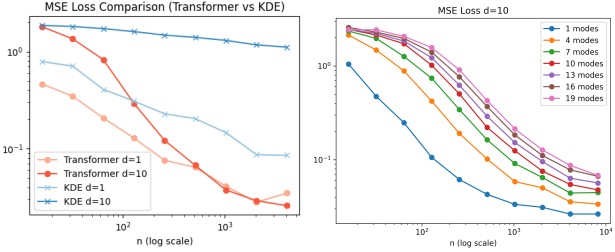

*Figure 4.* Left: MSE of score estimation in dimensions 1 and 10 on a 3-modal GMM using Transformer and KDE. The Transformer has excellent scaling in both dimension $d$ and the sample size $n$. Right: MSE of score estimation using the Transformer on GMMs with different numbers of modes. Despite being trained only on GMMs with 1-10 modes, and $n = 2048$, the model exhibits excellent generalization.

and compute the MSE score loss. Figure 3 demonstrates that even in dimension 2 and a KDE-favorable example of a unimodal Gaussian, our model performs better. The KDE score approximation is computed using

$$k(x_i, x_j) = \exp\left(-\frac{1}{2} \sum_{\ell=1}^{d} \frac{(x_i^\ell - x_j^\ell)^2}{h_\ell^2}\right),$$

$$\widehat{s}(x_i) = \frac{1}{h^2} \odot \left(\frac{\sum_j k(x_i, x_j) x_j}{\sum_j k(x_i, x_j)} - x_i\right),$$

where $h_\ell = \sigma_\ell n^{-1/(d+4)}$ is Scott's rule applied per coordinate with $\sigma_\ell$ the sample standard deviation along dimension $\ell$, and $\odot$ denotes element-wise multiplication.

For a quantitative comparison of score estimation against the KDE, we compute the Mean Squared Error (MSE) for dimensions 1 and 10 across several magnitudes of sample sizes $n$. See the left panel of Figure 4. The transformer model achieves much better accuracy than KDE even in dimension 1 and especially in dimension 10.

To judge the out-of-distribution generalization of our model, we compare the MSE loss on GMMs with 1-19 modes, while the training data consists of GMMs with 1-10 modes. The right panel of Figure 4 shows that the MSE loss is monotone and stable in the number of modes. This indicates good generalization. Additionally, we evaluate on two non-Gaussian distributions: the Laplace distribution, which has a sharper peak and heavier tails than a Gaussian, and the Student-$t$ distribution ($\nu = 3$), which has polynomial rather than exponential tail decay. Despite being trained only on GMMs, the model estimates the score successfully on both, as shown in Tables 2 and 3. Additionally, we can further improve the out-of-distribution performance using test-time training (TTT) (Sun et al., 2020; Gandelsman et al., 2022), using the consistency loss

$$\mathcal{L}_{\text{con}} = \frac{1}{n} \sum_{i=1}^{n} \|S(C, Q)_i - \nabla_{q_i} T(C, Q)_i\|_2^2,$$

where $T(C, Q)$ is the predicted log-density at queries $Q$ given context $C$, and $\nabla_{q_i} T$ – the gradient with respect to the query coordinate, with the context held fixed – is computed via automatic differentiation; at test time we set $C = \text{stopgrad}(X)$ and $Q = X$. Figure 5 and Table 3 show that as few as 4 TTT steps are sufficient to improve performance on both distributions.

## 4.2. High-Dimensional Estimation

To test scalability beyond the moderate dimensions ($d \leq 10$) used in the main experiments, we train a larger DiScoFormer in $d = 100$. Table 4 reports the results. DiScoFormer achieves a $6.5\times$ lower score MSE and a $37.5\times$ reduction in

*Table 2.* MSE of score estimation on the 2D Laplace distribution.

| $n$ | KDE | Transformer |
|---|---|---|
| 512 | 0.3810 | **0.3598** |
| 1024 | 0.3305 | **0.2992** |
| 2048 | 0.2990 | **0.2756** |
| 4096 | 0.2650 | **0.2597** |

*Table 3.* MSE of score estimation on the 2D Student-$t$ distribution ($\nu = 3$).

| $n$ | KDE | No TTT | TTT 4 | TTT 6 | TTT 8 |
|---|---|---|---|---|---|
| 128 | 0.1515 | 0.1980 | 0.1676 | 0.1569 | **0.1450** |
| 256 | 0.1206 | 0.1119 | 0.0956 | 0.0908 | **0.0897** |
| 512 | 0.0916 | 0.0574 | 0.0488 | **0.0485** | 0.0514 |
| 1024 | 0.0812 | 0.1023 | 0.0806 | 0.0768 | **0.0765** |

log-density MSE compared to the best KDE variant. This indicates that, in this high-dimensional regime, DiScoFormer scales far more gracefully than kernel methods, which are acutely affected by the curse of dimensionality.

### 4.3. Ablation: Effect of Whitening

To isolate the contribution of the whitening layer, we trained two identical DiScoFormer models that differed only in whether whitening was applied (in $d = 1$, whitening reduces to centering and variance scaling). Both were trained on GMMs whose location and scale parameters varied over multiple orders of magnitude. Table 5 reports the results. In-distribution, whitening provides a modest improvement but out-of-distribution — at scales well beyond the training range — the model without whitening catastrophically fails. This validates the theoretical motivation: whitening provides exact scale equivariance and approximate affine equivariance, enabling robust generalization to unseen scales.

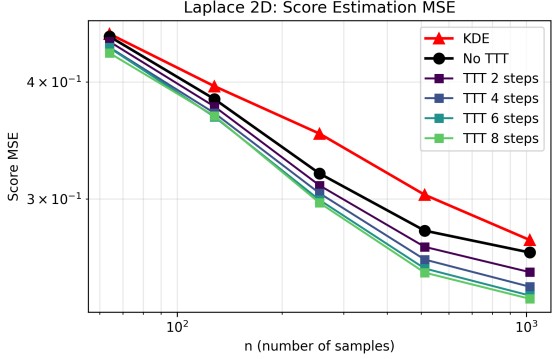

*Figure 5.* MSE of score estimation on the Laplace distribution. Test-time training (TTT) improves the out-of-distribution generalization.

*Table 4.* Score and log-density estimation in $d = 100$ on random 2-component diagonal-covariance GMMs ($n = 2048$ context, 256 queries). KDE baselines use two bandwidth strategies; "Oracle $h$" is the best fixed bandwidth found by grid search. DiScoFormer ($d_{\mathrm{model}} = 256$, 8 heads, 6 layers) is evaluated at 150k training steps.

| Method | Score MSE | Log-density MSE |
|---|---|---|
| Scott KDE | 1.155 | 967 |
| Oracle $h$ KDE | 1.090 | 781 |
| DiScoFormer | **0.167** | **20.8** |

*Table 5.* Ablation: effect of whitening on score and density estimation ($d = 1$). ID and OOD use different log-uniform meta-distributions over GMM location and scale parameters; OOD scales are well beyond the training range.

| | | Score MSE | Log-density MSE |
|---|---|---|---|
| ID | Whitening | **0.107** | **0.058** |
| | No whitening | 0.118 | 0.066 |
| OOD | Whitening | **0.020** | **0.123** |
| | No whitening | 1.136 | 1.593 |

### 4.4. Relative Fisher Information

Relative Fisher information compares two densities through their score fields:

$$\mathrm{I}(f\|g) = \mathbb{E}_{x \sim f}\|\nabla \log f(x) - \nabla \log g(x)\|^2.$$

Given samples $X = \{x_i\}_{i=1}^{n_x} \sim f$ and $Y = \{y_j\}_{j=1}^{n_y} \sim g$, DiScoFormer evaluates $\nabla \log g(x_i)$ by using $Y$ as context and $X$ as queries. This yields the Monte Carlo estimator

$$\widehat{\mathrm{I}}(f\|g) = \frac{1}{n_x} \sum_{i=1}^{n_x} \|S(X,X)_i - S(Y,X)_i\|^2,$$

where $S(C,Q)_i$ denotes the score estimate at query $q_i$ using context sample $C$. Figure 6 visualizes this. KL divergence can be estimated the same way using the log-density head, $\widehat{\mathrm{KL}}(f\|g) = \frac{1}{n_x} \sum_{i=1}^{n_x} \big( T(X,X)_i - T(Y,X)_i \big)$, where $T(C,Q)_i$ is the log-density estimate at query $q_i$ using context $C$.

### 4.5. Density Estimation via SD-KDE

Knowledge of the score $\nabla \log f$ can be exploited by score-debiased KDE (SD-KDE) (Epstein et al., 2025) to reduce the bias from $O(h^2)$ to $O(h^4)$. SD-KDE sharpens samples by moving them along the score before applying KDE: $X \mapsto X + \frac{h^2}{2} \nabla \log f(X)$, counteracting KDE's smoothing bias. In practice the true score is unavailable; we compare (i) plain Scott KDE, (ii) Emp-SD-KDE (Epstein et al., 2025) which uses Scott KDE as the score, (iii) SD-KDE with our Transformer-predicted score, (iv) SD-KDE with the autograd score $\nabla_y T(X, Y)$ of the Transformer's learned log-

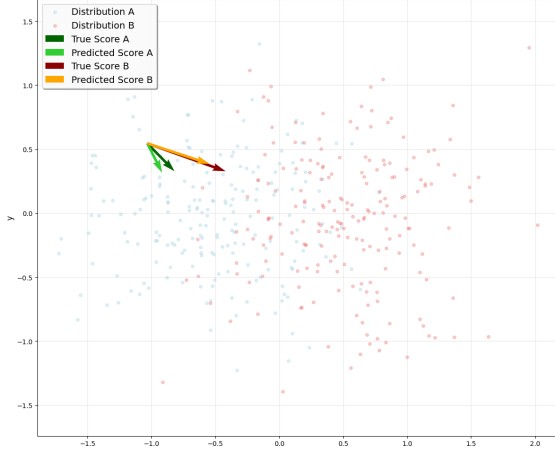

*Figure 6.* Computing relative Fisher information. Our model predicts $\nabla \log g(x_i)$ at query points $x_i$ via cross-attention with the samples $y_i \sim g$.

density, and (v) the Transformer's direct density estimate $\exp T(X, Y)$.

Figure 7 shows that the learned score substantially improves density recovery over plain KDE. In 1D bimodal GMMs (Figure 8), SD-KDE with the learned score best matches the truth. The direct density approach performs best at small $n$, but score-based methods scale better with $n$, as learning the score avoids the normalization constant $Z = \int_{\mathbb{R}^d} f$.

### 4.6. Estimation of entropy and Fisher information

Differential entropy $H(f) = -\mathbb{E}_f \log f$ and Fisher information $I(f) = \mathbb{E}_f \|\nabla \log f\|^2$ characterize the spread of a distribution. Plug-in estimates $\hat{H} = -\frac{1}{n} \sum_i T(X)_i$ and $\hat{I} = \frac{1}{n} \sum_i \|S(X)_i\|^2$ are computed directly from DiScoFormer's outputs. We compare against Scott KDE and Emp-SD-KDE on a 1D 3-mode GMM (entropy/Fisher computed by grid integration) and on $d = 10$ Gaussians with random mean/covariance (analytical formulas). Figure 10 shows that DiScoFormer beats KDE already in $d = 1$; in $d = 10$ the learned density wins for entropy, and the learned score wins for Fisher information.

### 4.7. Plasma simulation

The homogeneous Landau equation, a Fokker–Planck-type kinetic PDE used to simulate plasmas, can be rewritten (Carrillo et al., 2020) as a transport equation with velocity $v(x) = -\int A(x - y)(\nabla \log f(x) - \nabla \log f(y)) f(y) \, dy$ and solved by a particle method requiring an online score estimate. Score-Based Transport Modeling (Boffi & Vanden-Eijnden, 2023; Ilin et al., 2025; 2026; Ilin & Hu, 2026) fits a neural network via score matching at each step; this is accurate but expensive because of the on-the-fly retraining.

We reproduce experiments 5.3 and 5.4 of (Ilin et al., 2025) using DiScoFormer as a pre-trained score oracle, with *no* training during the simulation. Figure 11 shows that the Transformer matches the analytic covariance well, while the KDE-based solver struggles.

## 5. Conclusion

We introduced DiScoFormer, a permutation- and affine-equivariant Transformer that jointly estimates density and score from i.i.d. samples in a single forward pass. Unlike score matching, which learns $\mathbb{R}^d \to \mathbb{R}^d$ for one distribution, DiScoFormer learns a sequence-to-sequence operator that generalizes across distributions. We proved that a single self-attention layer, on squared-norm-lifted inputs, exactly represents the KDE score, and confirmed empirically that individual heads learn multi-scale kernel-like behaviors. DiScoFormer outperforms KDE across dimensions and sample sizes, and serves as a plug-in oracle for score-debiased KDE, entropy and Fisher information estimation, and Fokker–Planck-type PDEs.

**Limitations.** (i) *Training family.* We train exclusively on GMMs with up to 10 modes; Theorem B.2 bounds the gap to any smooth target by its best $K$-component GMM approximation, but distributions far from GMMs require retraining or finetuning. (ii) *Rotation equivariance.* Whitening provides exact equivariance under translation and scaling and only *approximate* rotation invariance via training augmentation. (iii) *Consistency.* Unlike KDE with a well-chosen bandwidth, DiScoFormer has no proven asymptotic guarantees; closing this gap is an interesting open question.

**Acknowledgments.** CPU and GPU computing was in part done using the UW Research Computing Club funded from the UW Student Technology Fee Committee. GPU computing was in part done on UWIT's GPU cluster Tillicum. We also gratefully acknowledge the use of CPU and GPU resources provided by the UW Department of Applied Mathematics.

## Impact Statement

This paper advances nonparametric density and score estimation, a foundational problem in statistics and machine learning. The methods we develop are general-purpose tools whose downstream uses—generative modeling, Bayesian inference, and the numerical solution of kinetic equations—inherit the broader societal consequences already associated with those areas; for example, improved score estimators contribute to generative models that carry well-documented risks around synthetic media and misuse. These consequences are not specific to our contribution, and we do not feel that any must be singled out here. We see no ethical

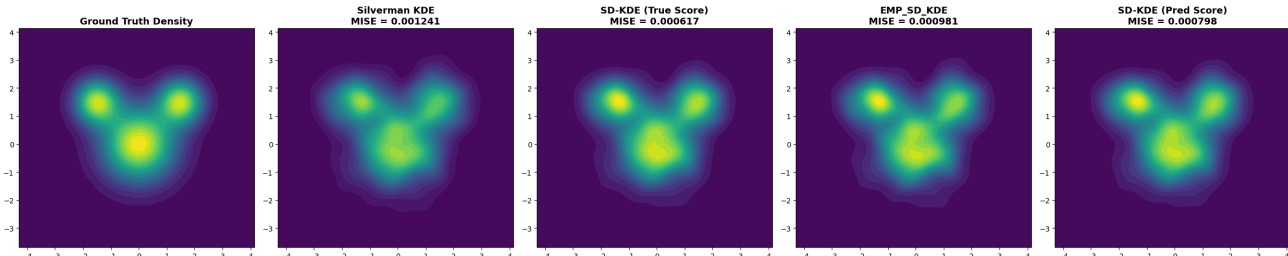

*Figure 7.* Qualitative comparison: True density, Scott KDE, Emp-SD-KDE, and SD-KDE with our learned score. Learned-score SD-KDE is visibly less biased.

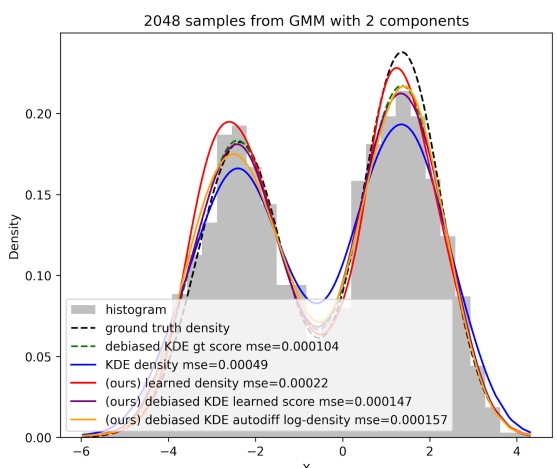

*Figure 8.* 1D bimodal GMM ($n = 2048$): SD-KDE with the learned score best approximates the density by MSE.

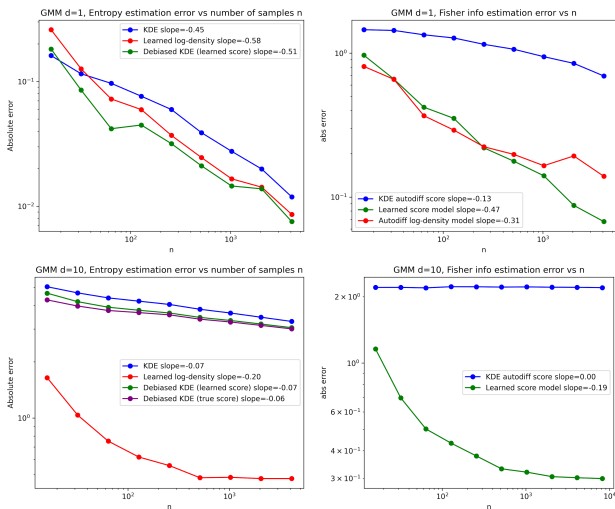

*Figure 10.* Comparison between the transformer model (learned) and Scott KDE and score-debiased KDE for estimation of differential entropy $H(f)$ and Fisher Information $I(f)$. Transformer's MSE is lower than that of the KDE approximation, even in dimension 1.

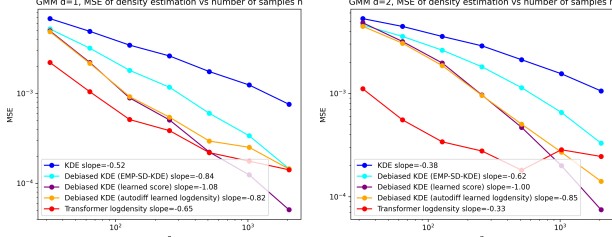

*Figure 9.* MSE of density estimation in 1D (left) and 2D (right). SD-KDE with our learned score and the Transformer model show the best scaling. The Transformer was trained only at $n = 2048$.

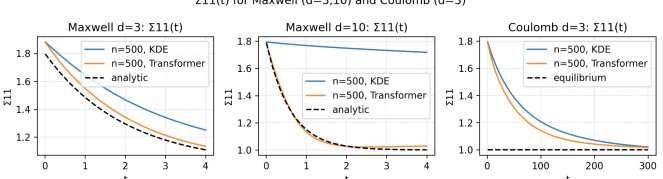

*Figure 11.* Comparison between the trained Transformer model and Scott KDE on the task of numerically solving the homogeneous Landau equation. We plot the first entry of the covariance matrix $\Sigma_{1,1}(t)$ of the numerical solutions and ground truth, when known. The left two panels use Maxwell collisions, while the last panel shows Coulomb collisions. The Transformer outperforms KDE, and is comparable in quality to SBTM in (Ilin et al., 2025).

concerns that are unique to this work beyond those generally
attendant to advancing the field of machine learning.

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

## A. Proofs

**Proposition A.1** (Permutation and affine equivariance of log-density and score evaluation). *Let $f : \mathbb{R}^d \to (0, \infty)$ be a density, and for*

$$X = \begin{pmatrix} x_1 \\ \vdots \\ x_n \end{pmatrix} \in \mathbb{R}^{n \times d}$$

*define*

$$T(X) := \begin{pmatrix} \log f(x_1) \\ \vdots \\ \log f(x_n) \end{pmatrix} \in \mathbb{R}^n, \qquad S(X) := \begin{pmatrix} \nabla \log f(x_1)^\top \\ \vdots \\ \nabla \log f(x_n)^\top \end{pmatrix} \in \mathbb{R}^{n \times d}.$$

*Let $P \in \mathbb{R}^{n \times n}$ be a permutation matrix, $A \in \mathbb{R}^{d \times d}$ be invertible, $\mu \in \mathbb{R}^d$, and $\mathbf{1} \in \mathbb{R}^n$ be the vector of ones. Then*

$$T(PXA + \mathbf{1}\mu^\top) = PT(X) - \log|\det A|\,\mathbf{1},$$
$$S(PXA + \mathbf{1}\mu^\top) = P\,S(X)\,A^{-\top}.$$

*Proof.* Write $X = (x_1^\top, \ldots, x_n^\top)^\top$ and let $\sigma$ be the permutation of $\{1, \ldots, n\}$ corresponding to $P$, i.e. $(PX)_{i\cdot} = x_{\sigma(i)}^\top$. Define

$$Y := PXA + \mathbf{1}\mu^\top \in \mathbb{R}^{n \times d}.$$

Then the $i$-th row of $Y$ is

$$y_i^\top = x_{\sigma(i)}^\top A + \mu^\top, \qquad \text{equivalently } y_i = A^\top x_{\sigma(i)} + \mu.$$

Consider the density obtained from $f$ by the affine change of variables $y = A^\top x + \mu$:

$$f^{(\mu, A)}(y) := |\det A|^{-1} f\big((A^\top)^{-1}(y - \mu)\big).$$

By construction,

$$f^{(\mu, A)}(y_i) = |\det A|^{-1} f\big(x_{\sigma(i)}\big).$$

Taking logarithms, $\log f^{(\mu, A)}(y_i) = \log f(x_{\sigma(i)}) - \log|\det A|$, so by the definition of $T$,

$$T(Y) = \begin{pmatrix} \log f^{(\mu, A)}(y_1) \\ \vdots \\ \log f^{(\mu, A)}(y_n) \end{pmatrix} = \begin{pmatrix} \log f(x_{\sigma(1)}) - \log|\det A| \\ \vdots \\ \log f(x_{\sigma(n)}) - \log|\det A| \end{pmatrix} = PT(X) - \log|\det A|\,\mathbf{1},$$

which proves the log-density equivariance.

For the score, differentiate $\log f^{(\mu, A)}$:

$$\log f^{(\mu, A)}(y) = -\log|\det A| + \log f\big((A^\top)^{-1}(y - \mu)\big).$$

By the chain rule,

$$\nabla_y \log f^{(\mu, A)}(y) = \big((A^\top)^{-1}\big)^\top \nabla_x \log f(x)\big|_{x = (A^\top)^{-1}(y - \mu)} = A^{-1} \nabla \log f\big((A^\top)^{-1}(y - \mu)\big).$$

Evaluating at $y_i = A^\top x_{\sigma(i)} + \mu$ gives

$$\nabla_y \log f^{(\mu, A)}(y_i) = A^{-1} \nabla \log f(x_{\sigma(i)}).$$

Thus, stacking the gradients as rows (so $S(Y) \in \mathbb{R}^{n \times d}$),

$$S(Y) = \begin{pmatrix} \nabla_y \log f^{(\mu, A)}(y_1)^\top \\ \vdots \\ \nabla_y \log f^{(\mu, A)}(y_n)^\top \end{pmatrix} = \begin{pmatrix} \nabla \log f(x_{\sigma(1)})^\top A^{-\top} \\ \vdots \\ \nabla \log f(x_{\sigma(n)})^\top A^{-\top} \end{pmatrix} = P\,S(X)\,A^{-\top},$$

using $\nabla_y \log f^{(\mu, A)}(y_i) = A^{-1} \nabla \log f(x_{\sigma(i)})$, which proves the score equivariance. $\qquad \square$

**Proof of Proposition 3.3 (Cross-attention computes reweighted Gaussian kernel smoothing)**

**Proposition A.2** (Restated). *Let $X \in \mathbb{R}^{n_x \times d}$ with rows $x_1, \ldots, x_{n_x}$ (context/keys) and $Y \in \mathbb{R}^{n_y \times d}$ with rows $y_1, \ldots, y_{n_y}$ (queries). For any positive semi-definite $B \in \mathbb{R}^{d \times d}$, define the cross-attention matrix $A_{ij} = \exp(y_i^\top B\, x_j) / \sum_{k=1}^{n_x} \exp(y_i^\top B\, x_k)$. Then*

$$A_{ij} = \frac{w_j\, \exp\!\left(-\frac{1}{2}\|y_i - x_j\|_B^2\right)}{\sum_{k=1}^{n_x} w_k\, \exp\!\left(-\frac{1}{2}\|y_i - x_k\|_B^2\right)},$$

*where $w_j = \exp\!\left(\frac{1}{2}\|x_j\|_B^2\right)$ and $\|z\|_B^2 = z^\top B\, z$.*

*Proof.* Since $B$ is positive semi-definite and symmetric, the polarization identity gives

$$\|y_i - x_j\|_B^2 = (y_i - x_j)^\top B(y_i - x_j) = \|y_i\|_B^2 + \|x_j\|_B^2 - 2\, y_i^\top B\, x_j.$$

Rearranging:

$$y_i^\top B\, x_j = \tfrac{1}{2}\|y_i\|_B^2 + \tfrac{1}{2}\|x_j\|_B^2 - \tfrac{1}{2}\|y_i - x_j\|_B^2.$$

Exponentiating both sides:

$$\exp(y_i^\top B\, x_j) = \exp\!\left(\tfrac{1}{2}\|y_i\|_B^2\right)\, \exp\!\left(\tfrac{1}{2}\|x_j\|_B^2\right)\, \exp\!\left(-\tfrac{1}{2}\|y_i - x_j\|_B^2\right).$$

Substituting into the softmax definition of $A_{ij}$, the factor $\exp(\frac{1}{2}\|y_i\|_B^2)$ depends only on the query index $i$ and appears in every term of the denominator sum. It therefore cancels between numerator and denominator, leaving

$$A_{ij} = \frac{w_j\, \exp\!\left(-\frac{1}{2}\|y_i - x_j\|_B^2\right)}{\sum_{k=1}^{n_x} w_k\, \exp\!\left(-\frac{1}{2}\|y_i - x_k\|_B^2\right)},$$

with $w_j = \exp(\frac{1}{2}\|x_j\|_B^2)$. Setting $Y = X$ recovers the self-attention statement. $\qquad\square$

**Proof of Proposition 3.5 (Cross-attention represents the KDE score and log-density)**

**Proposition A.3** (Restated). *Fix $h > 0$. Lift each context token to the key/value input $\tilde{x}_j = [\, x_j,\ \|x_j\|^2,\ 0_d\,] \in \mathbb{R}^{2d+1}$ and each query token to the residual-stream input*

$$H_i^{(0)} = [\, y_i,\ \|y_i\|^2,\ 0_d\,] \in \mathbb{R}^{2d+1}.$$

*Consider a bare, unmasked, residual cross-attention block with one head, exact row-wise softmax, affine query projection, linear key/value/output projections, no positional encodings, no layer normalization, no dropout, and no FFN. Then there are weights such that an affine readout of the block output and of the per-query log-normalizer $\ell_i := \log \sum_j \exp(q_i^\top k_j)$ gives, at every query $i$ and exactly in real arithmetic,*

$$\nabla_y \log \hat{f}_{h,X}(y_i) = \frac{1}{h^2}\left(\frac{\sum_j K_h(y_i, x_j) x_j}{\sum_j K_h(y_i, x_j)} - y_i\right), \qquad \log \hat{f}_{h,X}(y_i) = \ell_i - \frac{\|y_i\|^2}{2h^2} - \log n_x - \frac{d}{2}\log(2\pi h^2).$$

*If all projections are required to be strictly linear, the same construction works after adding a constant coordinate $1$ to every token, with $d_{\mathrm{model}} \geq 2d + 2$.*

*Proof.* Initialize the query residual stream as $H_i^{(0)} = [\, y_i,\ \|y_i\|^2,\ 0_d\,]$ and the context stream as $\tilde{x}_j = [\, x_j,\ \|x_j\|^2,\ 0_d\,]$, both in $\mathbb{R}^{2d+1}$; the squared-norm coordinates are supplied by the fixed lift, so no nonlinear computation occurs inside the block. All projections act identically on each token, preserving permutation equivariance in both streams.

**Query and key projections.** Let $d_k = d + 1$. Choose an affine query projection (on the query stream) and a linear key projection (on the context stream) so that, ignoring the standard attention scaling for the moment,

$$q_i = [\, y_i/h,\ 1\,], \qquad k_j = [\, x_j/h,\ -\|x_j\|^2/(2h^2)\,].$$

Both read off the lifted coordinates: $q_i$ takes $y_i/h$ from the first $d$ entries and the constant $1$ from the query bias (or, in the strictly linear variant, from the constant coordinate); $k_j$ takes $x_j/h$ from the first $d$ entries and $-\|x_j\|^2/(2h^2)$ by scaling the squared-norm coordinate. If the attention convention uses $q_i^\top k_j/\sqrt{d_k}$, multiply the query projection by $\sqrt{d_k}$ so that the scaled logit equals the displayed value.

**Attention weights are exact Gaussian kernels.** The attention logit is

$$a_{ij} := q_i^\top k_j = \frac{y_i^\top x_j}{h^2} - \frac{\|x_j\|^2}{2h^2} = -\frac{\|y_i - x_j\|^2}{2h^2} + \frac{\|y_i\|^2}{2h^2},$$

using $\|y_i - x_j\|^2 = \|y_i\|^2 + \|x_j\|^2 - 2y_i^\top x_j$. The last term is independent of $j$ and cancels in the row-wise softmax:

$$\alpha_{ij} = \text{softmax}_j(a_{ij}) = \frac{K_h(y_i, x_j)}{\sum_{k=1}^{n_x} K_h(y_i, x_k)}.$$

These are exactly the normalized Gaussian KDE weights—with no reweighting and no assumption on the norms $\|y_i\|, \|x_j\|$.

**Score from the value output.** Choose the value and output projections so that the attention aggregate is written only into the final $d$ scratch coordinates of the query stream,

$$\text{Attn}_i = [\,0,\ 0,\ m_i\,], \qquad m_i = \sum_{j=1}^{n_x} \alpha_{ij}\, x_j = \frac{\sum_j K_h(y_i, x_j)\, x_j}{\sum_j K_h(y_i, x_j)}.$$

After the residual connection (which carries the query stream), $H_i^{(1)} = H_i^{(0)} + \text{Attn}_i = [\,y_i,\ \|y_i\|^2,\ m_i\,]$, so the linear readout $[\,y_i,\ \|y_i\|^2,\ m_i\,] \mapsto (m_i - y_i)/h^2$ returns the exact KDE score $\nabla_y \log \hat{f}_{h,X}(y_i)$.

**Log-density from the normalizer.** The same softmax forms the per-query log-normalizer $\ell_i = \log \sum_j \exp(a_{ij})$ internally. Since $a_{ij} = \log K_h(y_i, x_j) + \|y_i\|^2/(2h^2)$ with the second term constant in $j$,

$$\sum_j \exp(a_{ij}) = \exp\left(\frac{\|y_i\|^2}{2h^2}\right) Z_i, \qquad Z_i := \sum_j K_h(y_i, x_j),$$

so $\ell_i = \|y_i\|^2/(2h^2) + \log Z_i$. The Gaussian KDE is $\hat{f}_{h,X}(y_i) = \frac{1}{n_x}(2\pi h^2)^{-d/2} Z_i$, hence

$$\log \hat{f}_{h,X}(y_i) = \log Z_i - \log n_x - \tfrac{d}{2}\log(2\pi h^2) = \ell_i - \frac{\|y_i\|^2}{2h^2} - \log n_x - \tfrac{d}{2}\log(2\pi h^2),$$

an affine readout of the two available quantities $\ell_i$ and $\|y_i\|^2$; exponentiating returns the density. (The $-\log n_x$ bias is constant for fixed $n_x$; for variable $n_x$ it is itself a log-normalizer $\log \sum_j e^0$, obtainable from an all-zero logit row.)

Every step is linear or affine except the fixed squared-norm feature and the softmax, so no approximation is involved. Setting $Y = X$ ($y_i = x_i$) recovers the self-attention score and density at the sample points. $\square$

*Remark* A.4 (Why the squared-norm feature is needed). The lifted coordinate supplies the key-only quadratic term of the Gaussian log-kernel. Indeed,

$$\log K_h(y_i, x_j) = [\,y_i/h,\ 1\,]^\top [\,x_j/h,\ -\|x_j\|^2/(2h^2)\,] - \frac{\|y_i\|^2}{2h^2},$$

and the final term depends only on the query index $i$, so it is invisible to the row-wise softmax. The lift therefore does *not* make the Gaussian RBF kernel a finite-dimensional inner-product (Mercer) kernel; it makes the attention *logits* equal to the Gaussian log-weights up to row-wise constants. Without it, affine $Q, K$ projections produce logits

$$(Ay_i + a)^\top (Bx_j + b) = y_i^\top A^\top B x_j + y_i^\top A^\top b + a^\top B x_j + a^\top b,$$

whose query-only parts ($y_i^\top A^\top b$ and $a^\top b$) cancel in the softmax, leaving a bilinear term in $(y_i, x_j)$ plus an affine key-only term in $x_j$; the key-only quadratic $-\|x_j\|^2/(2h^2)$ required for an exact Gaussian cannot arise. The squared-norm coordinate is the key-side scalar feature that supplies it; note that for the score it is needed only on the *context* tokens, since the query-side $\|y_i\|^2$ cancels in the softmax and the readout uses $y_i$ from the residual stream (the query lift $\|y_i\|^2$ is used only by the log-density readout above). Finally, among radial kernels $K(y, x) = \psi(\|y - x\|^2)$, the Gaussian is special: $\nabla_y \log K(y, x_j)$ is affine in $x_j$ exactly when $\log \psi$ is linear in $\|y - x_j\|^2$, which is what lets a linear value projection and linear readout recover the score from the normalized weights.

*Remark* A.5 (The density needs the normalizer, not just the value output). The log-density is not recoverable from the ordinary attention value output $m_i = \sum_j \alpha_{ij} x_j$ alone. For instance, in one dimension take context $x_1 = 0$, $x_2 = a$, $x_3 = -a$ and query $y = 0$: symmetry gives $m = 0$ for every $a > 0$, and the query's residual features $y, \|y\|^2$ are likewise independent of $a$; yet $Z(a) = 1 + 2e^{-a^2/(2h^2)}$, so $\hat{f}_{h,X}(y)$ varies with $a$. The normalized first moment thus carries the score but not the density scale, which is exactly the log-normalizer $\ell_i$. Moreover, at arbitrary query points the normalizer is not recoverable even from the *full* row of normalized weights: unlike the self-attention special case $Y = X$ – where the diagonal entry $\alpha_{ii} = K_h(x_i, x_i)/Z_i = 1/Z_i$ exposes $Z_i$ – a query $y_i \notin X$ has no self-token with $K_h(y_i, y_i) = 1$, so the softmax weights pin down $Z_i$ only up to the unknown scale by which they were normalized. The log-normalizer $\ell_i$ is therefore the clean object to expose for density estimation at arbitrary queries.

## B. Supplementary Theoretical Results

### B.1. Whitening as Adaptive Bandwidth Selection

The whitening step is equivalent to performing KDE with a data-adaptive bandwidth.

**Proposition B.1** (Whitening induces data-adaptive bandwidth). *Let $\hat{S} = X_c^\top X_c \succ 0$ and $\tilde{x}_i = \hat{S}^{-1/2}(x_i - \hat{\mu})$. Then the isotropic Gaussian KDE in the whitened space with scalar bandwidth $h$, pulled back to original coordinates, equals the anisotropic Gaussian KDE with full bandwidth matrix $H = h^2 \hat{S}$:*

$$\frac{1}{n} \sum_{i=1}^n \phi_{h^2 I}(\tilde{y} - \tilde{x}_i) \quad \xrightarrow{\text{change of variables}} \quad \frac{1}{n} \sum_{i=1}^n \phi_{h^2 \hat{S}}(y - x_i).$$

*Proof.* Under the affine map $y = \hat{S}^{1/2}\tilde{y} + \hat{\mu}$, the whitened KDE density transforms as $\hat{f}_h(y) = |\det \hat{S}^{-1/2}| \cdot \tilde{f}_h(\hat{S}^{-1/2}(y - \hat{\mu}))$. The exponent becomes $\|\hat{S}^{-1/2}(y - x_i)\|^2 = (y - x_i)^\top \hat{S}^{-1}(y - x_i)$, and the normalizing constant satisfies $|\det \hat{S}|^{-1/2} \cdot h^{-d} = |h^2 \hat{S}|^{-1/2}$, yielding $\hat{f}_h(y) = \frac{1}{n} \sum_{i=1}^n \phi_{h^2 \hat{S}}(y - x_i)$. $\square$

### B.2. Generalization from GMM Training to Non-GMM Targets

A natural question is why training exclusively on GMMs yields an estimator that generalizes to non-GMM distributions. The following result shows that the generalization gap depends on (i) how well a $K$-component GMM approximates the target in total variation, and (ii) how close the target functionals are — in $L^2$ for density estimation, or in Fisher divergence for score estimation. Crucially, the bound does *not* scale with the sequence length $N$, provided the trained network satisfies an $O(1/N)$ stability condition (stated precisely in Theorem B.2). We emphasize that this is an *assumption* on the learned model, not an automatic property of softmax attention, which can place $O(1)$ weight on a single token; in practice it is encouraged by the boundedness of the trained logits.

Throughout, the (population) *risk* of $T_\theta$ on a density $g$ is the expected squared error of its first-token output against the target functional,
$$\mathcal{R}(T_\theta, g) := \mathbb{E}_{X \sim g^N} \big[ \|T_\theta(X)_1 - f_g(x_1)\|^2 \big],$$
where $X = (x_1, \ldots, x_N)$ has i.i.d. rows $x_i \sim g$, $T_\theta(X)_1$ is the model output at the first token, and the target functional is $f_g(x) = g(x)$ for density estimation or $f_g(x) = \nabla \log g(x)$ for score estimation. The GMM training assumption is that $\mathcal{R}(T_\theta, p) \le \varepsilon$ for every $K$-component GMM $p$.

**Theorem B.2** (Generalization to non-GMM targets). *Let $T_\theta$ be a Transformer satisfying:*

- *Boundedness: outputs and targets bounded by $B$ on a compact domain $\mathcal{X}$.*

- *Stability: for sequences differing in one context token, $\|T_\theta(X)_1 - T_\theta(X^{(j)})_1\| \le L/N$ for $j \neq 1$.*

- *GMM training: population risk $\le \varepsilon$ on all $K$-component GMMs.*

*Let $g$ be a target density with best $K$-component GMM approximation $p^*$, with $D_{TV}(g, p^*) \le \delta_{TV}$. Then:*

(a) Density estimation: $\mathcal{R}(T_\theta, g) \le 2\varepsilon + 2\|g - p^*\|_{L^2(g)}^2 + C\,\delta_{TV}$,

(b) Score estimation: $\mathcal{R}(T_\theta, g) \le 2\varepsilon + 2\mathcal{I}(g\|p^*) + C\,\delta_{TV}$,

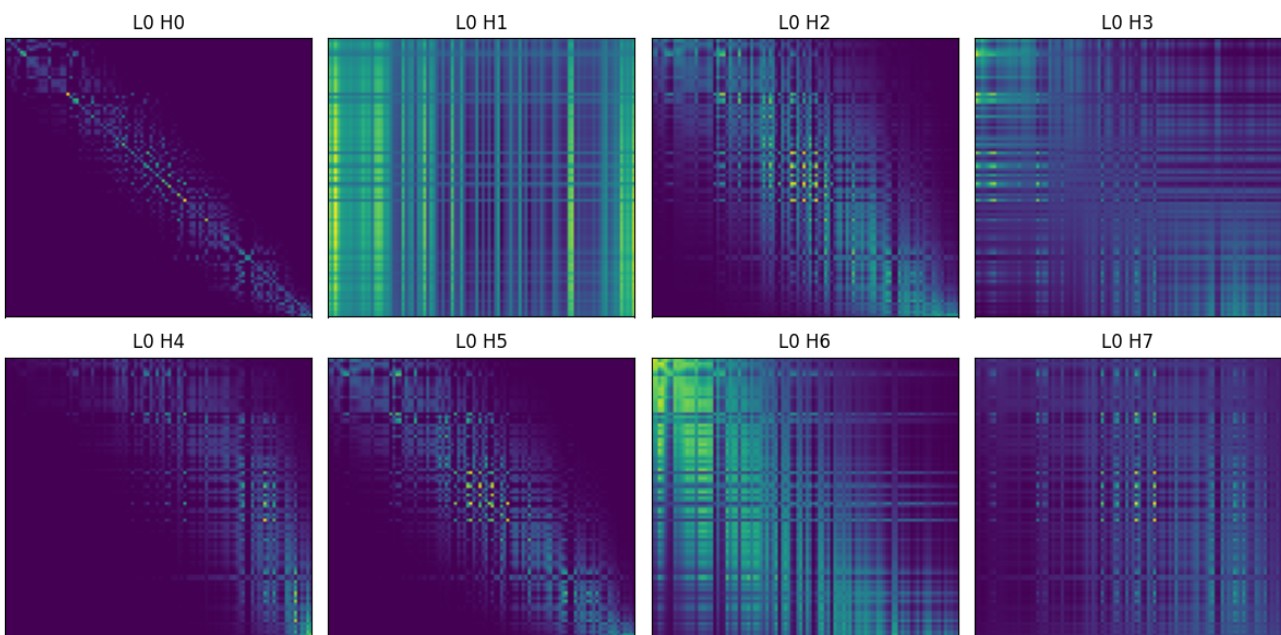

*Figure 12.* We visualize attention of the eight individual heads of layer 0 as a heatmap. Yellow color denotes higher attention weight and blue is lower. We choose the particle ordering so that nearby particles are close in the ordering. Heads 0, 2, and 5 specialize on nearby points, heads 1 and 6 specialize on far-away points, whereas heads 3, 4, and 7 attend in specific directions.

where $\mathcal{I}(g\|p^*) = \mathbb{E}_g[\|\nabla \log g - \nabla \log p^*\|^2]$ *is the relative Fisher information and* $C = 8B(B + L)$.

*Proof.* Decompose the risk via $(a+b)^2 \leq 2a^2+2b^2$: $\mathcal{R}(T_\theta, g) \leq 2\,\mathbb{E}_{g^N}[\|T_\theta(X)_1 - f_{p^*}(x_1)\|^2] + 2\,\mathbb{E}_g[\|f_{p^*}(x_1) - f_g(x_1)\|^2]$. The second term is $\|g - p^*\|^2_{L^2(g)}$ for density or $\mathcal{I}(g\|p^*)$ for score estimation.

For the first term, let $h(X) = \|T_\theta(X)_1 - f_{p^*}(x_1)\|^2$ (bounded by $4B^2$). We bound $|\mathbb{E}_{g^N}[h] - \mathbb{E}_{(p^*)^N}[h]|$ via a telescoping argument: couple $(x_j, x'_j)$ marginally so that $\Pr(x_j \neq x'_j) = D_{TV}(g, p^*)$. Changing the target token (position 1) costs at most $4B^2\,\delta_{TV}$ by the TV bound on bounded functions. Changing each context token $j \geq 2$ costs at most $\frac{2L \cdot 2B}{N}\,\delta_{TV}$ by the stability assumption. Summing over $N-1$ context tokens, the $N$ factors cancel:

$$|\mathbb{E}_{g^N}[h] - \mathbb{E}_{(p^*)^N}[h]| \leq 4B^2\,\delta_{TV} + 4BL\,\delta_{TV} = C\,\delta_{TV}/2.$$

Since $\mathbb{E}_{(p^*)^N}[h] \leq \varepsilon$ by assumption, we obtain $\mathbb{E}_{g^N}[h] \leq \varepsilon + C\,\delta_{TV}/2$, and the factor of 2 from the initial decomposition yields the stated bounds. $\square$

For smooth targets $g \in \mathcal{C}^s$, classical approximation theory gives $D_{TV}(g, p^*) = O(K^{-s/d})$ and $\|g - p^*\|_{L^2} = O(K^{-s/d})$, so the density bound (a) decays as $O(K^{-2s/d})$. For score estimation, assuming $p^*$ is bounded away from zero (so that $\nabla \log p^*$ is controlled), passing to $\nabla \log$ loses one derivative, $\|\nabla \log g - \nabla \log p^*\|_{L^2(g)} = O(K^{-(s-1)/d})$, and the Fisher divergence—a *squared* score error—satisfies $\mathcal{I}(g\|p^*) = O(K^{-2(s-1)/d})$. Thus, training on GMMs with sufficiently many components yields diminishing generalization error for any sufficiently smooth, non-degenerate target.

## C. Attention visualization

To interpret the outputs of the Transformer model we visualize the attention of the eight individual heads in layer 0 in two different ways. First, in Figure 12 we show the full attention matrices as heatmaps. We choose the particle ordering so that nearby particles are close in the ordering. Second, in Figure 13 we choose a random query point, marked with a red x, and color other points according to the strength of the query point's attention. We clearly observe an emergent behavior – heads specialize in different tasks. Head 1 specialized to look at far-away points, whereas Heads 0, 2, and 5 specialized to look at close- and mid-range interactions. Finally, Heads 3, 4, 6, and 7 specialized to look in specific directions. This emergent behavior further links the multi-head Transformer to kernel-based methods, but with multi-scale learned kernels.

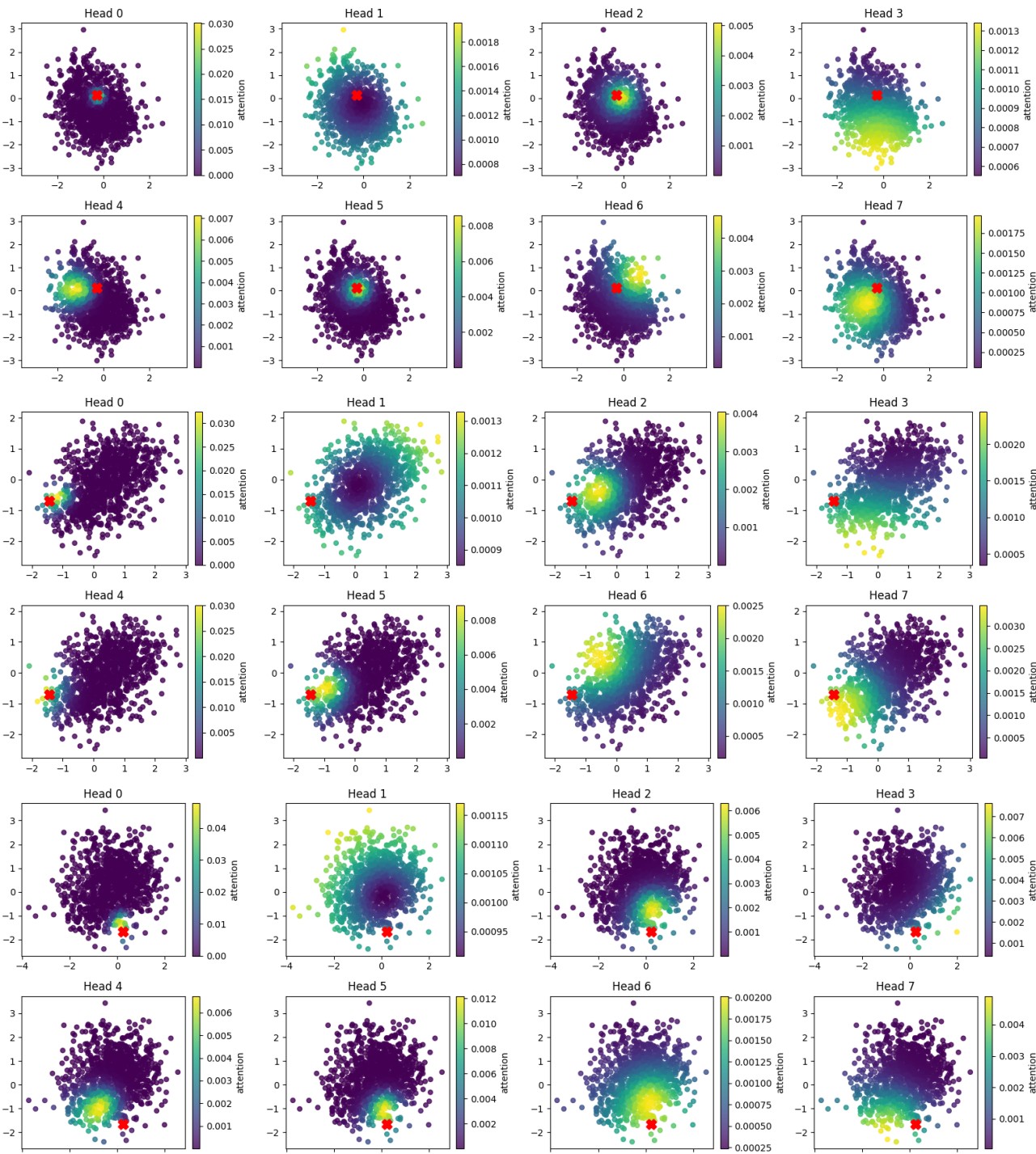

*Figure 13.* We visualize attention of the eight individual heads of layer 0 as a scatter plot. We choose a random query point, marked with a red x. Yellow color denotes higher attention weight and blue is lower. Heads 0, 2, and 5 specialize on nearby points, head 1 specializes on far-away points, whereas heads 3, 4, 6, and 7 attend in specific directions.

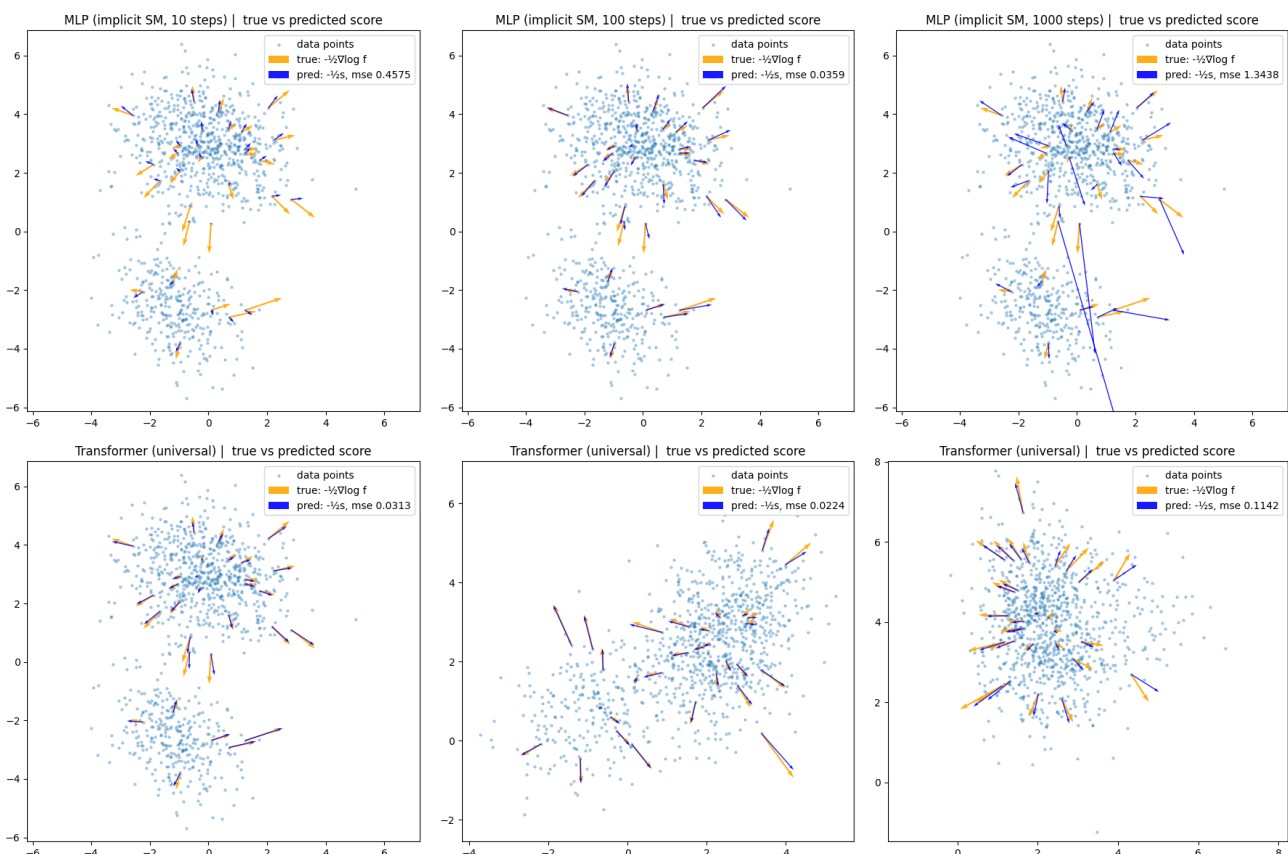

*Figure 14.* Comparison of sliced score matching and our transformer model. The transformer model can be used without retraining and does not suffer from overfitting. We plot the negated score for ease of visualization.

## D. Comparison with the Score Matching Loss

The Hyvrinen score matching loss (Hyvärinen, 2005) estimates the data score without access to the true density by minimizing

$$\mathcal{L}_{\mathrm{SM}}(\theta) = \mathbb{E}_{x \sim f(x)}\big[\| s_\theta(x) \|^2 + 2\,\nabla_x \cdot s_\theta(x)\big],$$

which corresponds to the Fisher divergence $\mathbb{E}_f[\|s_\theta(x) - \nabla_x \log f(x)\|^2]$ up to a constant.

To avoid explicitly computing the divergence term, one can apply a finite-difference denoising trick, replacing $\nabla \cdot s_\theta(x)$ with a stochastic estimator

$$\nabla \cdot s_\theta(x) \ \approx \ \mathbb{E}_{z \sim \mathcal{N}(0,I)} \left[ \frac{z^\top \big(s_\theta(x + \alpha z) - s_\theta(x - \alpha z)\big)}{2\alpha} \right],$$

yielding a practical divergence-free approximation used in many implementations of score matching.

In Figure 14 we compare the performance of the score matching loss compared to the proposed Transformer model. The Transformer model is superior in two respects. First, it does not need to be retrained on new samples. Second, the score matching loss is easy to under- or over-fit, as demonstrated in the figure by training for 10, 100 and 1000 steps.

## E. Runtime comparison

While KDE is known to fail in moderate and high dimensions, it is thought to be very fast. We compare the wall-clock runtime of KDE score estimation (Scott's-rule bandwidth) and the Transformer model we propose. The Transformer model contains $800,000$ learned parameters. We use both KDE and the Transformer model on a single L40S GPU with 48GB of memory. Both methods are $O(n^2)$ asymptotically due to pairwise computations (kernel evaluations for KDE, attention

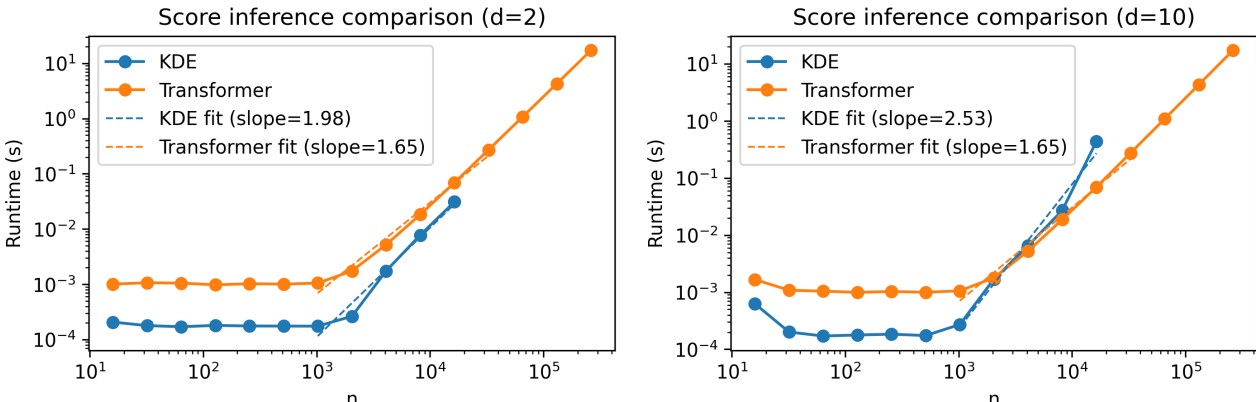

*Figure 15.* Runtime comparison between KDE and the Transformer model in 2 and 10 dimensions. Both are $O(n^2)$ asymptotically, but empirically the Transformer scales better and has improved memory efficiency. KDE encounters an OOM error at $n = 2^{15}$.

for Transformers). However, in practice the observed scaling differs: Figure 15 shows that while KDE is faster for small sample size $n \leq 2048$, it becomes slower after this point, especially in higher dimension. Additionally, the naive KDE implementation runs out of GPU memory after $n = 32,768$, while the Transformer benefits from the highly optimized attention kernels in modern deep learning frameworks, which are difficult to match with custom KDE implementations in practice.

