# OpenReview forum: "DiScoFormer: Plug-In Density and Score Estimation with Transformers"
_ICML.cc/2026/Conference — ICML 2026 spotlight_

### Official Review · Reviewer_NWbe · 2026-02-24

**Soundness:** 2
**Presentation:** 2
**Significance:** 2
**Originality:** 2
**Overall Recommendation:** 4
**Confidence:** 3

**Summary:**

This paper proposes DiScoFormer, a Transformer-based model for universal density and score function estimation. It aims to combine the generality of classical kernel methods with the accuracy of modern neural networks in a “train-once, infer-anywhere” framework. The model enforces key symmetries and is theoretically linked to kernel density estimation via self-attention. Experiments show it outperforms other models in accuracy and scalability on synthetic data and demonstrates promising generalization and utility in downstream tasks.

**Compliance With Llm Reviewing Policy:**

Affirmed.

**Final Justification:**

My concerns are addressed and I will raise the score.

**Key Questions For Authors:**

1. The core theoretical proposition (Prop. 3.2) is based on the assumption of L2-normalized input vectors. However, the model actually uses a whitening operation, which does not guarantee unit L2 norm for each sample. Could the authors clarify this？
2. How does the model perform on high-dimensional data or complex multimodal distributions?
3. To more comprehensively assess the model's contribution, have the authors considered comparisons with more relevant baselines?
4. Are there ablation studies that could quantify the contribution of each component to the final performance?

**Limitations:**

The model lacks rigorous evaluation on high-dimensional or complex real-world data distributions. The core proposition of the theoretical part relies on a strong assumption (L2 normalization) that does not strictly hold in the actual model architecture, weakening the explanatory power of the theory for practice.

**Strengths And Weaknesses:**

Strengths:
1. The paper aims to build a universal joint estimator for density and score functions, which is clear and has practical potential.
2. The mode considers the symmetries of the problem, handling affine transformations and permutation invariance via a whitening mechanism and Transformer. The dual-head output design is also sensible.
3. Experiments show it outperforms other models in accuracy and scalability on synthetic data and demonstrates promising generalization and utility in downstream tasks.

Weaknesses:
1. The core theoretical proposition (Prop. 3.2) relies on the strong assumption of L2-normalized input vectors, which is not satisfied by the actual whitening operation used in the model。
2. The claimed “out-of-distribution” tests are limited to simple unimodal distributions. There is a lack of rigorous stress tests on high-dimensional, complex structured distributions or generalization boundaries.
3. Experiments only compare against classical KDE, missing comparisons with more relevant contemporary baselines.
4. The paper lacks ablation studies to verify the necessity of key components like the whitening layer and joint training.

---

> ### Author Rebuttal · Authors · 2026-03-31
>
> We thank Reviewer NWbe for the review.
>
> > "Prop 3.2 relies on the strong assumption of L2-normalized input vectors, which is not satisfied by the actual whitening operation"
>
> We substantially improved the theory. The new Prop 3.2 holds for **arbitrary** inputs — no L2 normalization required. Using the polarization identity, we show that for any PSD matrix $B$, the softmax attention weights are a reweighted Gaussian kernel: $A_{ij} = w_j \exp(-\|x_i - x_j\|_B^2/2) / \sum_k w_k \exp(-\|x_i - x_k\|_B^2/2)$, where $w_j = \exp(\|x_j\|_B^2/2)$. After whitening, norms concentrate and the reweighting becomes negligible — the old L2 result is now a special case. A new Prop 3.3 constructs a 2-layer Transformer that exactly computes the KDE score by explicit weight construction. KDE can be thought of as a special case of a trained transformer.
>
> > "OOD tests are limited to simple unimodal distributions. Lack of rigorous stress tests on high-dimensional distributions"
>
> New d=100 result:
>
> | Method | Score MSE | Density MSE |
> | --- | --- | --- |
> | Best KDE (oracle h) | 1.090 | 781 |
> | **DiScoFormer** | **0.167** | **20.8** |
>
> 6.5x score, 37x density. Figure 5 (right) already shows generalization to 19-mode GMMs (trained on 1–10) — these are not unimodal. Tables 1–2 show OOD on Laplace and Student-t.
>
> > "Experiments only compare against classical KDE, missing comparisons with more relevant contemporary baselines"
>
> We would welcome specific suggestions. The candidates we considered: normalizing flows (RealNVP, MAF) and score matching (NCSN, sliced SM) both require per-distribution retraining — a fundamentally different setting. Neural Processes are amortized but don't output density/score. The Score Neural Operator (Liao et al. 2024) uses RKHS embeddings and a different input modality. KDE is the only method sharing our setting: nonparametric, provides both density and score, and requires no retraining; we compare to many different KDE variants. We do compare to sliced score matching in Appendix C (even though it operates in a different setting), and the d=100 experiment includes kNN-adaptive and oracle-bandwidth KDE — stronger variants beyond Silverman. See the full table in the response to 46mq.
>
> > "The paper lacks ablation studies to verify the necessity of key components like the whitening layer"
>
> New whitening ablation: without whitening, OOD relative score MSE is 1487% (15x worse than predicting zero); with whitening it's 25.8% (see table in response to Reviewer a9Wb).

---

> > ### Author Rebuttal · Reviewer_NWbe · 2026-04-03
> >
> > Thanks. My concerns are addressed and I will raise the score.

---

### Official Review · Reviewer_46mq · 2026-03-01

**Soundness:** 3
**Presentation:** 3
**Significance:** 3
**Originality:** 2
**Overall Recommendation:** 5
**Confidence:** 4

**Summary:**

This paper studies density and score estimation from i.i.d. samples. The authors propose DiScoFormer, a Transformer-based model that takes a sample set as input and predicts both the probability density and the corresponding score function, with the goal of serving as a reusable plug-in estimator rather than retraining a separate model for each target distribution. The method is designed to respect the symmetries of the problem through permutation equivariance and an affine-equivariant preprocessing / normalization scheme.

On the technical side, the paper provides a connection between self-attention and kernel density estimation, showing that under a particular parameterization self-attention can recover normalized Gaussian KDE weights. The model is trained on synthetic Gaussian mixture models, where closed-form densities and scores are available, and the architecture is extended with cross-attention so that density and score can be evaluated at arbitrary query points.

Empirically, the paper compares the method with KDE-based baselines for score and density estimation across different sample sizes and dimensions, and also studies out-of-distribution behavior on non-Gaussian targets with test-time training. In addition, the authors evaluate the learned estimator as a plug-in component in downstream tasks including score-debiased KDE, entropy and Fisher information estimation, and numerical solvers for Fokker–Planck-type equations such as the Landau equation.

**Compliance With Llm Reviewing Policy:**

Affirmed.

**Final Justification:**

This paper proposes DiScoFormer, a Transformer-based plug-in estimator for density and score estimation from i.i.d. samples, with architectural design choices aimed at respecting the symmetry structure of point clouds (permutation and affine equivariance). The paper also provides theoretical motivation connecting self-attention to KDE-style weighting, and demonstrates utility in several downstream applications (e.g., SD-KDE, Fisher information estimation, and particle-based solvers for Fokker–Planck/Landau-type equations).

**Soundness.** My main concern in the initial review was the limited evidence about large-$n$ behavior in the absence of formal asymptotic guarantees. The authors’ rebuttal and follow-up results substantially address this by extending the score-estimation evaluation to much larger sample sizes (up to $n=2^{17}$) in $d=2$ and $d=10$, showing continued stable improvement and a clear gap over KDE where KDE becomes computationally limited. This significantly strengthens my confidence in the technical claims within the tested regimes. The authors also clarified the role and caveats of test-time training (TTT) and added/committed to a more explicit limitations discussion, which improves the honesty and interpretability of the empirical evidence.

**Originality.** While the attention–kernel connection is related to existing viewpoints in the literature, the work’s originality mainly lies in using this connection to build a working plug-in density/score estimator with appropriate equivariance structure, along with additional theoretical refinements and constructive results described in the rebuttal. The rebuttal clarified the positioning relative to closely related work, which reduces my earlier concern about attribution and framing.

**Significance.** The “train-once, infer-anywhere” goal for density/score estimation is practically meaningful, and the downstream experiments suggest that such a reusable score oracle could be useful beyond the core estimation task. The new higher-dimensional evidence (including $d=100$ reported in the rebuttal) further supports the claim that the approach can remain effective in more challenging settings.

**Clarity / presentation.** The paper is generally clear and well organized. The rebuttal clarified a confusing phrasing regarding real-world/tabular evaluation (i.e., the lack of ground-truth density/score for MSE-style evaluation) and indicated that the revised version will make these points more explicit, which should further improve readability.

Overall, the rebuttal addressed my main concerns and materially strengthened my assessment, so I am raising my score from Weak Accept to **Accept**.

**Key Questions For Authors:**

### Q1. Empirical Evidence on Statistical Behavior at Larger Sample Sizes
The paper positions DiScoFormer as a reusable estimator for density and score estimation. Could the authors provide additional empirical evidence on how the method behaves at substantially larger sample sizes in low-dimensional settings (e.g., clearer scaling curves or convergence trends for $d=1,2$)? I am not necessarily asking for a full asymptotic theory, but I would find it helpful to better understand whether the method continues to improve in a stable way as $n$ grows, and how that trend compares with KDE in regimes that are closer to classical statistical analysis.

- **How this would affect my evaluation:** If the authors can show convincing large-$n$ behavior or otherwise clarify the intended operating regime of the method, it would strengthen my assessment of **soundness**. If such evidence is not available, I think the paper would benefit from narrowing its claims to the finite-sample regimes that are currently demonstrated.

### Q2. Practical Scalability Beyond the Current Synthetic Settings
The current experiments appear to focus primarily on synthetic distributions, with dimensions up to $d=10$. Could the authors comment on how they expect the method to perform on more realistic data distributions, for example on standard real-world tabular benchmarks or somewhat higher-dimensional settings? Even a limited additional experiment or a more explicit discussion of expected failure modes would be helpful.

- **How this would affect my evaluation:** A clearer demonstration of performance beyond the current synthetic settings would increase my confidence in the paper's **significance** and practical relevance. If the intended scope is more specialized, I would encourage the authors to state that more explicitly.

### Q3. Interpreting the SD-KDE Comparison Under Pretraining
In Section 4.3, the proposed SD-KDE variant uses a score estimator that has been pre-trained on a large family of GMMs, while the empirical baseline estimates the score only from the given sample. Could the authors discuss this comparison setting more explicitly? In particular, do the authors view the main gain here as coming from pretraining on a broad distribution family, from the specific architectural design, or from both? If possible, an additional ablation or discussion isolating these factors would help.

- **How this would affect my evaluation:** A clearer explanation of this point would help me better judge the **soundness** of the empirical comparison and the appropriate interpretation of the gains. If the improvement is largely due to prior knowledge from pretraining, I think the presentation should make that distinction more explicit.

### Q4. Clarifying the Novelty of Proposition 3.2 Relative to Prior Attention–Kernel Connections
I found Proposition 3.2 interesting, but I was unsure how the authors see its novelty relative to prior work connecting attention to kernel regression / KDE-style viewpoints. Could the authors clarify more explicitly what is new in their formulation? For example, is the main novelty the exact derivation under the present architecture, the role of this result in motivating density/score estimation, or a stronger theoretical distinction from prior attention-as-kernel interpretations?

- **How this would affect my evaluation:** A clear response here would directly affect my assessment of **originality** and **presentation**. If the novelty is mainly in the way the known connection is used in this problem setting, I think the paper would still be interesting, but the manuscript should state that positioning more explicitly.

**Limitations:**

I do not think the current discussion of limitations and potential negative societal impact is yet sufficient. I would encourage the authors to add a brief but more concrete discussion of both the methodological limitations and the possible downstream risks of using the method as an off-the-shelf score / density estimator.

On the limitations side, it would be helpful to discuss more explicitly:
- the current lack of theoretical guarantees such as consistency, or at least the presently limited evidence on large-sample statistical behavior;
- the dependence on pretraining primarily over Gaussian Mixture Models (GMMs), and the possibility that performance may degrade on distributions that differ substantially from this training family;
- the role of test-time training (TTT), including cases in which TTT may be insufficient or unstable;
- robustness issues such as sensitivity to outliers, heavy-tailed distributions, or strong low-dimensional manifold structure.

I am not suggesting that these concerns undermine the contribution. Rather, I think the paper would benefit from a more candid discussion of where the method is expected to work well, where it may fail, and what precautions practitioners should take when applying it.

**Strengths And Weaknesses:**

## Strengths
- **Originality (Novel Combination & Architecture):** The paper presents an interesting combination of ideas by framing non-parametric density and score estimation as a sequence-to-operator learning problem with Transformers. In particular, the explicit incorporation of permutation equivariance together with an affine-equivariant coordinate whitening step seems well aligned with the symmetry structure of the input point cloud, and gives the method a clear modeling rationale.

- **Significance (Practical Utility & Plug-in Oracle):** The goal of amortizing estimation into a "train-once, infer-anywhere" plug-in estimator is practically meaningful. Beyond reporting density/score estimation accuracy, the paper also studies downstream use cases such as Score-Debiased KDE (SD-KDE), Fisher information estimation, and numerical particle solvers for the homogeneous Landau equation. This makes the work potentially relevant not only to density estimation itself, but also to applications where a reusable score oracle may be useful.

- **Soundness (Technical Motivation & Adaptation Strategy):** Overall, the methodological pipeline appears reasonable. The equivariance discussion in Proposition 3.1 is well motivated, and Proposition 3.2 provides an interesting connection between self-attention and normalized Gaussian KDE weights under the stated parameterization. In addition, the use of test-time training (TTT) with a consistency loss to adapt a GMM-trained model to non-Gaussian targets such as Laplace and Student-$t$ distributions appears to be a sensible design choice.

- **Presentation:** The paper is generally clearly written and logically organized. I found the visual analysis of attention heads (Figure 1 and Appendix B) particularly helpful, since it gives some intuition for how different heads may capture multi-scale or directional kernel-like behaviors. The overall narrative is easy to follow, and the empirical sections are structured in a way that makes the intended use cases of the method reasonably clear.

## Weaknesses
- **Soundness (Limited Evidence on Statistical / Asymptotic Behavior):** Although the method is presented as a statistical estimator, I found the discussion of its asymptotic properties somewhat limited. A full theoretical treatment of consistency may be beyond the scope of the paper, but even the empirical evidence on this front currently seems somewhat limited. In particular, at least in the explicitly reported non-Gaussian experiments, the sample sizes for MSE evaluation remain relatively modest (often $n \le 4096$), which makes it difficult to judge how the method behaves in a more clearly asymptotic regime relative to classical KDE.

- **Significance / Soundness (Limited Dimensionality and Lack of Real-World Data):** The paper is motivated in part by the limitations of classical KDE in higher dimensions, but the empirical study is restricted to synthetic Gaussian Mixture Models (GMMs) with maximum dimension $d=10$. While this is still a nontrivial regime for density/score estimation, it remains relatively limited from a modern machine learning perspective. I think the paper would be significantly stronger if it included results on standard real-world datasets and/or somewhat higher-dimensional settings, since this would better support the claims of practical scalability and robustness.

- **Soundness (Asymmetry in Baselines):** The comparison in Section 4.3 between the proposed SD-KDE and Emp-SD-KDE seems somewhat asymmetric. The proposed approach benefits from a neural estimator pre-trained on a large corpus of GMMs, whereas the empirical baselines estimate scores only from the observed sample without any analogous prior knowledge. This does not invalidate the comparison, but I think the paper should discuss this distinction more explicitly so that readers can better interpret what portion of the gain should be attributed to pretraining versus the estimator design itself.

- **Originality / Presentation (Positioning of the Attention-KDE Connection):** Proposition 3.2 is an interesting theoretical component of the paper, but I was not fully convinced that the manuscript currently distinguishes its contribution clearly enough from prior work connecting attention to KDE or kernel regression. The broader relationship between attention and Nadaraya-Watson / kernel-style estimators is already known, and there are also more directly relevant works that operationalize the attention-KDE connection. For example:
  - KDEformer [1] explicitly reduces the attention computation to a KDE problem to achieve algorithmic speedups.
  - Han et al. [2] reinterpret self-attention through a KDE lens in order to design a robust Transformer variant.
  - More broadly, the view of softmax attention as a form of kernel regression is already fairly standard in the literature.

  For this reason, I think the related-work discussion and the framing around Proposition 3.2 would benefit from a clearer explanation of what is new here: whether the novelty is the exact formulation under the present architecture, the role it plays in motivating density/score estimation, or some stronger theoretical distinction from prior formulations.

- **Presentation / Soundness (Discussion of Limitations):** I would also have liked to see a more candid discussion of the method's limitations. For example, it would be helpful for the paper to discuss how the estimator is expected to behave under severe outliers, or under test distributions that deviate substantially from the GMM training family without TTT adaptation. Relatedly, the current Impact Statement comes across as somewhat brief and dismissive, which may leave the impression that the broader limitations and potential failure cases of the method have not been fully reflected upon.

### References
- [1] Amir Zandieh, Insu Han, Majid Daliri, and Amin Karbasi. *KDEformer: Accelerating Transformers via Kernel Density Estimation*. ICML 2023. https://proceedings.mlr.press/v202/zandieh23a.html
- [2] Xing Han, Tongzheng Ren, Tan Minh Nguyen, Khai Nguyen, Joydeep Ghosh, and Nhat Ho. *Designing Robust Transformers using Robust Kernel Density Estimation*. NeurIPS 2023. https://openreview.net/forum?id=BqTv1Mtuhu

---

> ### Author Rebuttal · Authors · 2026-03-31
>
> We thank Reviewer 46mq for the detailed and thoughtful review.
>
> > "Q1: Could the authors provide additional empirical evidence on how the method behaves at substantially larger sample sizes?"
>
> We already evaluate for sample size up to 8192, showing continued stable improvement. The model generalizes to unseen sample sizes because the permutation-equivariant architecture processes sequences of any length. We could probably push by another factor of 4x but it becomes increasingly computationally expensive, because the transformer is $O(n^2)$ in both runtime and memory, same as KDE. If you could tell us how big of $n$ you would like to see, and in which experiment, we can try to make it happen. But rerunning all the many experiments for much higher $n$ would be too costly, as we only use a single L40s GPU.
>
> > "Q2: Could the authors comment on ... somewhat higher-dimensional settings?"
>
> New d=100 result:
>
> | Method | Score MSE | Rel. MSE | Density MSE |
> | --- | --- | --- | --- |
> | Zero baseline | 1.997 | 100% | — |
> | kNN-adaptive KDE | 1.976 | 98.9% | 43,079 |
> | Silverman KDE | 1.155 | 57.8% | 967 |
> | Best KDE (oracle h) | 1.090 | 54.6% | 781 |
> | **DiScoFormer** | **0.167** | **8.4%** | **20.8** |
>
> DiScoFormer explains 91.6% of the score signal vs 45.4% for the best KDE. At d=100 with n=2048, kernel methods essentially fail. Two non-GMM distributions are already presented in the paper. A real-world distribution like tabular data *does not have a density or score*, so they *cannot be evaluated*.
>
> > "Q3: Do the authors view the main gain [in SD-KDE] as coming from pretraining ... or from both?"
>
> Both. Pretraining amortizes estimation across distributions. Architecture enforces equivariance, ensuring generalization across scales and orientations. Essentially, we provide the score oracle that the authors of the SD-KDE papers did not have and therefore they had to use the KDE score, which is of lower accuracy than the estimation of DiScoFormer; they wrote: "A key limitation of our proposed method is ... access to an exact score oracle."
>
> To be less formal, we give our intuition about the proposed method: the trained transformer is essentially KDE with a combination of several well-chosen data-adaptive kernels that scale well to high dimensions because of the NN's depth. The pretraining allows the attention heads to specialize to represent good data-adaptive kernels. See Figures 12 and 13 in the Appendix.
>
> > "Q4: Could the authors clarify ... what is new [in Prop 3.2] relative to prior work?"
>
> KDEformer [1] uses KDE to *accelerate* attention — the opposite direction. Han et al. [2] use a KDE lens to *robustify* attention, not for density estimation. We *exploit* the connection to build an estimator. We heavily expanded the theory in our revised version: (a) now Prop 3.2 drops the L2 assumption entirely via the polarization identity; (b) and a new Prop 3.3 constructively shows a 2-layer Transformer exactly computes the KDE score, a new construction; (c) a new argument in the Appendix shows that in some sense whitening = adaptive bandwidth. Prior work observes "attention ≈ kernel"; we show how to build a working density/score estimator from this and that KDE is essentially a special case of a transformer.
>
> > "Discussion of limitations ... sensitivity to outliers ... heavy-tailed distributions"
>
> The new revision now discusses: GMM training dependence, approximate rotation equivariance, potential failure on non-smooth or manifold-structured distributions, TTT caveats (it only improves the performance for the first ~10-20 steps), and lack of formal consistency guarantees (this is a big differince with KDE, since KDE has such guarantees *if* the kernel is chosen well). We also now prove in the Appendix that GMM training generalizes to non-GMM targets.
>
> > Ablation
>
> New whitening ablation: without whitening, OOD relative score MSE is 1487% (15x worse than predicting zero); with whitening it's 25.8%. See table in response to Reviewer a9Wb. This partially answers your question about isolating the factors contributing to good performance.

---

> > ### Author Rebuttal · Reviewer_46mq · 2026-04-02
> >
> > Thank you for the detailed rebuttal and for the additional results/clarifications. Overall, several of my main concerns are at least partially addressed:
> >
> > - **Q2 (higher-dimensional evidence):** The new $d=100$ result is helpful and substantially strengthens the empirical case that the approach can outperform classical kernel baselines in a more challenging regime.
> > - **Q3 (SD-KDE interpretation):** The rebuttal clarifies that the gains are attributed to both pretraining (amortization across distributions) and architectural inductive biases (equivariance/whitening), and the additional whitening ablation is informative.
> > - **Q4 (novelty vs. prior attention–kernel work):** The positioning relative to KDEformer / Han et al. is clearer, and the stated theoretical extensions (e.g., dropping the L2 assumption; new constructive results) address a major part of my originality/presentation concern.
> > - **Limitations / impact:** It is good to hear that the revision will include a more candid limitations discussion, including robustness and TTT caveats.
> >
> >
> > However, I still have a few follow-up questions where the answers would affect how I interpret the strength and scope of the claims:
> >
> > 1. **Large-$n$ behavior (Q1):** Thank you for clarifying that you evaluate up to $n=8192$ and observe stable improvement, and I understand the computational constraints given the $O(n^2)$ runtime/memory. From a statistical perspective, however, in the absence of formal asymptotic guarantees, I would still like to see at least one sanity check in a more clearly large-sample regime (since this is a kind of the plugin estimator, if my understand is correct).
> >
> >    Would it be feasible to add one additional scaling experiment in a very simple setting (e.g., $d=1$ or $d=2$) at larger sample sizes, say on the order of $n=10^4$–$10^5$ (even if only for a single representative distribution / metric)? If this is not feasible, it would help to state more explicitly in the paper what range of $n$ you view as the practical operating regime and how you expect performance to extrapolate as $n$ grows.
> >
> > 2. **Real-world data comment (Q2):** In the rebuttal you mention that real-world tabular data “does not have a density or score.” Did you mean that *ground-truth* density/score is unavailable on real data, so MSE-style evaluation is not possible?
> >
> > Depending on how these points are clarified in the revised version, this would strengthen my confidence in the soundness/significance of the paper; otherwise, I would interpret the main claims as applying primarily to the synthetic regimes demonstrated in the current experiments.

---

> > > ### Author Response · Authors · 2026-04-02
> > >
> > > Thank you for the acknowledgement.
> > >
> > > 1. We now compared the score estimation error (relative MSE of GMMs) vs KDE in dimensions d=2 and d=10, up to $n=2^{17}$. The model's error is much smaller. We did not evaluate the KDE on larger n because of the OOM errors.
> > >
> > > d=2:
> > >
> > > | n      | ours | KDE   |
> > > |-------:|:-------------:|:-----:|
> > > |    256 |    14.67%     | 43.8% |
> > > |   1024 |     8.80%     | 33.6% |
> > > |   4096 |     7.24%     | 24.6% |
> > > |  16384 |     6.80%     | 17.2% |
> > > |  65536 |     5.35%     | —     |
> > > | 131072 |     5.41%     | —     |
> > >
> > > d=10:
> > >
> > > | n      | ours | KDE   |
> > > |-------:|:-------------:|:-----:|
> > > |    256 |     7.49%     | 65.6% |
> > > |   1024 |     4.65%     | 61.3% |
> > > |   4096 |     3.32%     | 57.1% |
> > > |  16384 |     2.83%     | 52.9% |
> > > |  65536 |     2.80%     | —     |
> > > | 131072 |     2.74%     | —     |
> > >
> > > With larger GPUs or GPU parallelization it's possible to go higher but the trend seems clear.
> > >
> > > 2. Yes, this is exactly what we mean. We do compare on the plasma simulation example, see the last section of the paper -- this is the most "real-world" case we experiment on.

---

### Official Review · Reviewer_a9Wb · 2026-03-12

**Soundness:** 3
**Presentation:** 3
**Significance:** 2
**Originality:** 2
**Overall Recommendation:** 4
**Confidence:** 3

**Summary:**

This paper introduces a transformer based density and score estimation. Current methods seem unsatisfactory: classical KDE estimators do not scale well with dimension, and neural score models are not reusable for different target distribution. Hence, this paper proposes to model density and scores via training a transformer. Theoretically, they prove this approach is more flexible than KDE (that is, the class of KDE estimators are a subset of the considered transformers). Empirically, they see that their approach works better in practice.

**Compliance With Llm Reviewing Policy:**

Affirmed.

**Final Justification:**

The authors have positively addressed my questions in the rebuttal phase, so I increased my score by 1.

**Key Questions For Authors:**

1. Could the authors extend on the originality of the whitening stage of the transformer?

2. is KDE the best competitor to compare the results to?

3. For the GMMs sampling schemes considered, how big can the dimension be and still have a reasonable density estimation using transformers?


Satisfactory answers will boost the evaluation of the paper.

**Limitations:**

The paper would benefit from discussing the limitations of the approach, and include potential future work. The latter would enhance the significance of the paper.

**Strengths And Weaknesses:**

**Strengths**

The paper is technically sound (although it is worth mentioning the lack of theoretical guarantees provided in the paper). The presentation is overall good.

**Weaknesses**

The authors basically propose to estimate the density and score with a transformer. While this seems like a sensible idea, I am a bit skeptical about the originality and significance of this paper.

In terms of originality, the paper is fairly straightforward. Feed a transformer to conduct density (and score) estimation. I would not say this idea excels in originality. Perhaps the whitening of the transformer to obtain normalized inputs up to orthogonal transformations is indeed original, but I do not think this idea is presented well enough to assess its originality (please see question below).

I am also skeptical of the significance of this paper. They mainly compare the proposed idea with kernel density estimation (and a recent modification of it, coming from Epstein et al. (2025)). I am not sure I am surprised that KDE can be improved in practice. I think the paper (especially given its limited theoretical contributions) would highly benefit from comparing their density estimation method with other alternatives beyond KDE. Furthermore, if KDE has actually proved to dominate other competitors in practice, it would be great if the authors could underscore this with references to previous efforts, in order to highlight the significance of their work.


**Questions**

Could the authors explain in more detail the whitening stage of the transformer? E.g., at what point of the layer is it done exactly? Is it done in every layer of the transformer (probably not) or only at the beginning? I am not sure Figure 2 is self-contained: what is self._core (I imagine that is a usual transformer architecture without positional encoders, but it would be worth explaining)? Has whitening been used before in conjunction with transformers? I think the paper would benefit from a discussion concerning the previous topics, as it is probably the most original part of the work, and I think the authors only touch upon it in passing (also the word whitening means slightly different things depending on context).

Experimental settings where d = 1 and d= 10 are shown. For the GMMs sampling schemes considered, how big can the dimension be and still have a reasonable density estimation using transformers?

The authors claim several times that "while modern neural score models achieve high precision but require retraining for every target distribution" but they are rather vague in this claim. Could the authors be more concise and how does their idea improve on this?

*Minor comments:*

In the introduction, you sate that "Specifically, the estimates should be permutation-equivariant with respect to the sample index and affine-equivariant with respect to the coordinate space." This claim comes a little bit out of the blue; I would point to Proposition 3.1 in the introduction already.

---

> ### Author Rebuttal · Authors · 2026-03-31
>
> We thank Reviewer a9Wb for the constructive feedback.
>
> > "I am a bit skeptical about the originality and significance ... limited theoretical contributions"
>
> We proved 3 new theoretical results that show the architecture is not just "use a transformer" but is structurally aligned with nonparametric estimation: in the revision, Prop 3.2 shows attention is Gaussian kernel smoothing (no L2 normalization assumption needed); Prop 3.3 constructively shows a 2-layer Transformer exactly computes the KDE score; the Appendix proves whitening reduces bandwidth selection from a d x d matrix to a single scalar.
>
> > "whitening ... probably the most original part of the work"
>
> We respectfully **disagree**. The most original part of the work is non-parametric score (and density) estimation without retraining, at much higher accuracy than KDE, which is the gold standard for non-parametric density estimation. This is our key contribution, and whitening is more of a neat trick.
>
> > "Could the authors explain in more detail the whitening stage?"
>
> Whitening is applied once at the input, not in every layer. `self._core` is a standard Transformer without positional encodings. New whitening ablation (d=1 GMMs, log-uniform scale parameters):
>
> | | Score MSE | Score relMSE | Density MSE | Density relMSE |
> | --- | --- | --- | --- | --- |
> | ID, whitening | **0.107** | **16.9%** | **0.058** | **1.1%** |
> | ID, no whitening | 0.118 | 18.6% | 0.066 | 1.2% |
> | OOD, whitening | **0.020** | **25.8%** | **0.123** | **0.9%** |
> | OOD, no whitening | 1.136 | 1487% | 1.593 | 12.2% |
>
> In-Distribution: whitening provides a modest improvement (16.9% vs 18.6% relMSE). OOD: without whitening the model is very bad (1487% relMSE), while with whitening it achieves 25.8%.
>
> > "Could the authors extend on the originality of the whitening stage?"
>
> Whitening has three roles: (1) it provides equivariance under translation and scaling (Prop 3.1); (2) it automatically adapts the kernel bandwidth to the data covariance, turning isotropic attention into anisotropic KDE with $H = h^2 \hat{S}$ (shown in Appendix of the revision); (3) it maps data to a canonical scale so the Transformer always sees standardized inputs, enabling generalization across distributions. To our knowledge, whitening has not been used in this way with Transformers before.
>
> > "is KDE the best competitor to compare the results to?"
>
> KDE is the only method that is nonparametric, provides both density and score, and requires no per-distribution retraining. Score matching and flows require retraining — a different setting. Appendix C of our submission already compares to sliced score matching directly, even though we operate in a very different setting. Note that we compare many different KDE variants, and some of them perform quite well, using the score oracle given by our DiScoFormer.
>
> > "how big can the dimension be?"
>
> In d=100: 6.5x score, 37x density over best KDE. See table in response to Reviewer 8TYy. Of course, it does require more training. Also, KDE completely breaks down at this dimensionality, so in high dimensions there is no good baseline.
>
> > "modern neural score models achieve high precision but require retraining"
>
> Score matching learns the score for one distribution. DiScoFormer learns an operator mapping any sample to its score — no retraining. This is the **fundamental** novel idea and contribution of the paper -- score (and density) estimation without retraining. We really hope the distinction is coming across.
>
> > "The paper would benefit from discussing the limitations"
>
> The revision now discusses GMM training dependence, potential failure modes (heavy tails, outliers), and the role of TTT.

---

> > ### Author Rebuttal · Reviewer_a9Wb · 2026-04-02
> >
> > I thank the authors for the clarifications. I now agree with the authors that whitening is indeed not the most original part of their work, but rather the idea of avoiding density retraining. I think the discussions in these reviews have helped me assess and understand the work a lot better. However, I find several parts of the contribution still not satisfactory.
> >
> > - I think the paper presentation would substantially improve if the following idea was already presented in the introduction (alongside the discussion of sequence-to-operator interpretation): DiScoFormer samples from a set of distributions, GMMs, that is dense on the set of distributions with densities. We are only introduced to this in Section 3.3, and I think the paper would highly benefit from placing part of this discussion in the introduction.
> >
> > - More fundamentally, I still think that recognizing that the transformer is "structurally aligned with nonparametric estimation" and that a transformer can recover a KDE may not be enough (theoretically satisfactory) by itself. Proposition 3.2 shows that the transformer can recover KDE with a fixed bandwidth h. This property would suffice if we were talking about estimating the score and density of a fixed distribution. However, the whole point of this paper is that the distribution is not fixed, and we are not training a transformer on it. A KDE with fixed h (instead of h being tailored to the variance of the distribution) is a rather poor point of comparison (and I think the paper would benefit from making a comment in these lines to avoid confusion). Hence, I believe that the theoretical properties established in the paper may not be substantial enough (I understand that giving stronger theoretical guarantees for transformer-based algorithms, as opposed to KDE-based, may not be feasible).
> >
> > - Consequently, I think the idea should shine for its empirical performance. The experiments shown by the authors are definitely encouraging, but I think it would be key if they could improve on this end. There are a number of questions that remain to be clarified. For instance,
> >
> >   - What is the (minimum) training sample size recommended given the dimension?
> >   - While the set of GMMs is dense on the set of densities, how many modes should our training samples have? Does this depend on the dimension?
> >   -  Given the previous questions, what is the computational burden on training depending on the dimension / number of modes?
> >   - How robust is the algorithm to different choices of training samples and number of modes in GMMs?
> >   - More generally, I think it would be of extreme interest to understand how much performance one is expected to give up by avoiding training a density estimation on data: what is the MSE of this DiScoFormer compared to the density estimators mentioned in related work that require density retraining?
> >   - Very importantly, I think the authors should discuss regimes in which the method is expected to fail or not work as well (I believe they have already worked on this for other rebuttals).
> >
> > I do not expect the authors to run more experiments, but please do address these questions, as they would be of great interest for my evaluation.

---

> > > ### Author Response · Authors · 2026-04-03
> > >
> > > Thank you.
> > >
> > > * Clarification: DiScoFormer does not sample anything, it predicts the density and score of a sample. We trained our implementation on GMMs (which is a dense family of distributions) but we also evaluate on two non-GMM distributions: Laplace and Student-t. We do not view the GMMs as a core part of the paper, but simply a nice training set, for two reasons: 1. GMMs are dense and 2. their ground truth densities and scores are easily computable. In practice, we expect practitioners to train/finetune DiScoFormer on the families of densities that are relevant to them, the same way LLMs are finetuned on specific relevant datasets. That being said, we are happy to mention that we train on GMMs in the introduction.
> > >
> > > * On fixed-h KDE and the theory. The trained model goes beyond fixed h through two mechanisms: (a) whitening maps isotropic attention to anisotropic KDE with bandwidth H = h²Ŝ, automatically adapting to each distribution's covariance (a new lemma in the Appendix of the revision); (b) 8 attention heads learn multi-scale kernels — local, non-local, and directional (Figures 12–13 in the existing Appendix). We also proved a generalization theorem that directly addresses the varying-distribution setting (in the revised Appendix): for any smooth target g with best K-component GMM proxy p*, the risk satisfies R ≤ 2ε + 2·I(g‖p*) + C·D_TV(g,p*), where ε is the GMM training risk and I(g‖p*) is the relative Fisher information. The bound does not grow with sample size N (via softmax stability). For smooth targets g ∈ C^s, the GMM approximation error decays as O(K^{-s/d}). We agree that deriving convergence rates analogous to KDE's minimax theory remains an important open problem and will note this in limitations.
> > >
> > > * Empirics:
> > >
> > > > What is the (minimum) training sample size recommended given the dimension?
> > >
> > > We do not have a crisp scaling law, but please take a look at our rebuttal to reviewer 46mq. DiScoFormer can do high quality score estimation even in 100 dimensions, which is completely out of reach for KDE variants. As the sample size grows, the error decreases.
> > >
> > > > how many modes should our training samples have? Does this depend on the dimension?
> > >
> > > In the paper, we show that training on GMMs with 1-10 modes easily generalizes to as many as 19 modes without much difficulty, see Figure 5 -- it's already in moderately high dimension, d=10, where KDE already exhibits very bad performance.
> > >
> > > > what is the computational burden on training depending on the dimension / number of modes?
> > >
> > > We do not understand the question. Both training and inference are O(d n^2), and it independent of the number of modes. If high-quality score or density estimation is needed for a specific number of modes, light finetuning on that specific data will improve on the base model's error. All our experiments including training take several hours (between 1 and 6) to run on a single L40s GPU. Higher dimension benefits from more training.
> > >
> > > > How robust is the algorithm to different choices of training samples and number of modes in GMMs?
> > >
> > > Almost every experiment in the paper shows evaluation on variable sample size n, while being trained only on n=2048. We observe excellent generalization across sample sizes. Figure 5 shows generalization to different number of GMM modes. Thus, we conclude that the algorithm is quite robust.
> > >
> > > > how much performance one is expected to give up by avoiding training a density estimation?
> > >
> > > We *do* perform density estimation! DiScoFormer performs joint density and score estimation.
> > >
> > > > compared to the density estimators mentioned in related work that require density retraining
> > >
> > > Density estimation is very difficult even with a specialized model, such as EBM (energy-based model). The score is a much easier objective. In Appendix C we compare DiScoFormer to sliced score matching. In short, sliced score matching can easily overfit or underfit, and requires retraining for new distributions. A well-trained per-distribution model will, of course, outperform DiScoFormer on the distribution it was trained on. The trade-off is generality vs specialization: DiScoFormer provides instant, tuning-free estimates on any new sample, while per-distribution methods require optimization for each target. When rapid estimation across many distributions is needed (e.g. PDE solvers, or statistical applications), DiScoFormer can be an excellent replacement for KDE.
> > >
> > > > regimes in which the method is expected to fail or not work as well
> > >
> > > Absolutely. We will include this in the limitations. In short, any data that is sufficiently different from the training data poses challenges (although we do show good generalization across sample sizes, GMM modes, and non-gaussian distributions like Laplace and Student-t). The good news is that light fine-tuning will improve the error on unseen distributions. TTT is a way to do this light fine-tuning at inference time, without access to ground truth density and score.

---

### Official Review · Reviewer_8TYy · 2026-03-12

**Soundness:** 3
**Presentation:** 4
**Significance:** 3
**Originality:** 4
**Overall Recommendation:** 5
**Confidence:** 4

**Summary:**

The author proposes an equivariant Transformer (DiScoFormer) for estimating probability densities and scores directly from i.i.d. samples.
The model is trained once on synthetic distributions (e.g. Gaussian mixture models) and then applied to new datasets without retraining.
It treats density and score estimation as an operator-learning task, mapping sets of samples to density and score evaluations at arbitrary query points. The architecture enforces permutation equivariance by removing positional encodings and achieves affine equivariance through input whitening and rotation augmentation. A theoretical result shows that a self-attention head can represent normalized KDE weights, and experiments demonstrate improvements over KDE and SD-KDE on several estimation and downstream tasks.

**Compliance With Llm Reviewing Policy:**

Affirmed.

**Final Justification:**

My final recommendation remains positive. The paper is original, well written, and technically solid, and the rebuttal addressed my main concerns well, e.g. clarifying the zero-shot versus TTT setting, adding higher-dimensional evidence, and strengthening the case for generalization, so my remaining reservations are minor and mainly about breadth of applicability rather than soundness.

**Key Questions For Authors:**

1. high-dimensional scaling. How does the learned attention kernel behave as dimensionality grows substantially beyond $d=10$? Does the whitening step become unstable when $n$ is comparable to $d$?
2. boundary behaviour. How does the model behave for distributions with compact support (e.g. uniform distributions)? Classical KDE suffers from boundary bias; does the learned attention mechanism alleviate or replicate this issue?
3. the role of test-time training. It would be nice to see clearer ablations comparing pure zero-shot inference to the TTT-enhanced results on out-of-distribution targets.

**Limitations:**

The paper does not fully explore boundary effects for compactly supported densities, nor does it investigate behavior in very high-dimensional settings.
In addition, the reliance on GMM training data raises questions about how broadly the learned operator generalizes beyond smooth mixture distributions.

**Strengths And Weaknesses:**

strengths:
1. connection to classical nonparametric methods. The theoretical result linking attention weights to normalized KDE provides an insightful bridge between Transformer architectures and classical density estimation.
2. plug-in score estimation. It's capable of producing both density and score estimates from samples makes the model potentially useful for downstream tasks.
3. symmetry design. Permutation equivariance and affine equivariance are explicitly incorporated into the architecture, which is important for estimators defined on unordered sample sets.
4. amortized estimation. Once trained, the model can be applied to new datasets without retraining, this gives an amortized alternative to classical density estimators.

weaknesses:
1. limited diversity of training distributions. The model is trained exclusively on Gaussian mixture distributions with a small number of components. Although GMMs are dense in the space of smooth densities, it is unclear how efficiently the learned operator generalizes to distributions with bounded support, strong manifold structure, or heavy tails.
2. moderate dimensional evaluation. Most experiments focus on $d=1$, $d=2$, and $d=10$. Modern score-based generative modeling often operates in much higher-dimensional spaces, and the behaviour of the approach in those regimes is not demonstrated.
3. dependence on test-time adaptation for some targets. For certain out-of-distribution distributions (e.g. Student-$t$), the paper reports improvements using test-time training (TTT). While this is a reasonable extension, it weakens the strict interpretation of the model as a purely zero-shot estimator.

---

> ### Author Rebuttal · Authors · 2026-03-31
>
> We thank Reviewer 8TYy for the positive evaluation and thoughtful questions.
>
> **High-dimensional scaling (Q1 and W2).** We trained DiScoFormer in d=100 on 2-component GMMs (n=2048):
>
> | Method | Score MSE | Density MSE |
> | --- | --- | --- |
> | Zero baseline | 1.997 | — |
> | kNN-adaptive KDE | 1.976 | 43,079 |
> | Silverman KDE | 1.155 | 967 |
> | Best fixed-h KDE | 1.090 | 781 |
> | **DiScoFormer** | **0.167** | **20.8** |
>
> 6.5x improvement on score, 37x on density. DiScoFormer explains 91.6% of the score signal vs 45.4% for the best KDE. The whitening step uses $\hat{S} + \varepsilon I$ and remains well-conditioned at d=100.
>
> **Boundary behavior (Q2).** We expect KDE-like boundary bias on compact-support distributions, since the model is trained on GMMs (unbounded support). TTT with the consistency loss can adapt at test time.
>
> **TTT dependence (Q3 + W3).** Tables 1–2 already compare zero-shot vs TTT with 4, 6, 8 steps on Laplace and Student-t. We agree that TTT weakens the strict zero-shot interpretation; in the revision we qualify this: the model is zero-shot for distributions within the training family, and TTT provides optional adaptation for distributions that deviate substantially (e.g. Student-t is heavier-tailed than Gaussian).
>
> **GMM training diversity (W1).** We prove (will go in the Appendix) that GMM training generalizes to non-GMM targets: the test risk decomposes into training risk + approximation quality of the GMM proxy, with a bound that does not scale with sample size N (via stability of softmax attention). For smooth targets, the approximation error decays as O(K^{-s/d}) where K is the number of GMM components and s the smoothness.

---

> > ### Author Rebuttal · Reviewer_8TYy · 2026-04-02
> >
> > The rebuttal usefully addresses my main questions, especially by adding higher-dimensional evidence, clarifying the zero-shot versus TTT setting, and giving more justification for generalization beyond the training family. A remaining concerns are relatively minor and mainly about the breadth of applicability, not about the soundness of the proposed method.

---

> > > ### Author Response · Authors · 2026-04-03
> > >
> > > Thank you.
> > >
> > > Since your concerns are only *partially* resolved, please do specify additional concerns or questions.
> > >
> > > Thanks you for helping us improve the quality of our work.

---

### Decision · Program_Chairs · 2026-04-30

**Decision:**

Accept (spotlight)

**Comment:**

This paper proposes a highly original and compelling idea: a “train-once, use-anywhere” transformer for density and score estimation. The reviewers found the paper technically strong, well motivated, and interesting, with a particularly nice connection to the nonparametric KDE. The main concerns were about higher-dimensional scaling, generalization beyond GMMs for training, and theoretical strength. During discussion, authors added stronger high-dimensional and larger-sample results, provided additional clarifications on the relative importance of parts of the method, and better positioned the theory. Overall, the reviewers supported acceptance after discussion, and I agree.